# Ageing compromises mouse thymus function and remodels epithelial cell differentiation

Jeanette Baran-Gale[1†], Michael D Morgan[2,3†], Stefano Maio[4,5], Fatima Dhalla[4,5], Irene Calvo-Asensio[6], Mary E Deadman[4,5], Adam E Handel[4], Ashley Maynard[7], Steven Chen[7], Foad Green[7], Rene V Sit[7], Norma F Neff[7], Spyros Darmanis[7], Weilun Tan[7], Andy P May[7], John C Marioni[2,3,8], Chris P Ponting[1*], Georg A Holländer[4,5,6,9*]

[1]MRC Human Genetics Unit, University of Edinburgh, Edinburgh, United Kingdom; [2]Wellcome Sanger Institute, Wellcome Genome Campus, Hinxton, United Kingdom; [3]Cancer Research United Kingdom - Cambridge Institute, Li Ka Shing Centre, University of Cambridge, Cambridge, United Kingdom; [4]Weatherall Institute of Molecular Medicine, University of Oxford, Oxford, United Kingdom; [5]Department of Paediatrics, University of Oxford, Cancer Research, Oxford, United Kingdom; [6]Department of Biomedicine, University of Basel, and University Children's Hospital, Basel, Switzerland; [7]Chan Zuckerberg Biohub, San Francisco, United States; [8]EMBL-EBI, Wellcome Genome Campus, Hinxton, United Kingdom; [9]Department of Biosystems Science and Engineering, ETH Zurich, Basel, Switzerland

*For correspondence:
Chris.Ponting@igmm.ed.ac.uk
(CPP);
georg.hollander@paediatrics.ox.
ac.uk (GAH)

†These authors contributed equally to this work

**Abstract** Ageing is characterised by cellular senescence, leading to imbalanced tissue maintenance, cell death and compromised organ function. This is first observed in the thymus, the primary lymphoid organ that generates and selects T cells. However, the molecular and cellular mechanisms underpinning these ageing processes remain unclear. Here, we show that mouse ageing leads to less efficient T cell selection, decreased self-antigen representation and increased T cell receptor repertoire diversity. Using a combination of single-cell RNA-seq and lineage-tracing, we find that progenitor cells are the principal targets of ageing, whereas the function of individual mature thymic epithelial cells is compromised only modestly. Specifically, an early-life precursor cell population, retained in the mouse cortex postnatally, is virtually extinguished at puberty. Concomitantly, a medullary precursor cell quiesces, thereby impairing maintenance of the medullary epithelium. Thus, ageing disrupts thymic progenitor differentiation and impairs the core immunological functions of the thymus.

## Introduction

Ageing compromises the function of vital organs via alterations of cell type composition and function (*López-Otín et al., 2013*). The ageing process is characterised by an upregulation of immune-system-associated pathways, referred to as inflamm-ageing, which is a conserved feature across tissues and species (*Benayoun et al., 2019*). Ageing of the immune system first manifests as a dramatic involution of the thymus. This is the primary lymphoid organ that generates and selects a stock of immunocompetent T cells displaying an antigen receptor repertoire purged of pathogenic 'Self' specificities, a process known as negative selection, yet still able to react to injurious 'Non-Self' antigens (*Palmer, 2013*). The thymus is composed of two morphological compartments that convey different functions: development of thymocytes and negative selection against self-reactive antigens are both

initiated in the cortex before being completed in the medulla (*Abramson and Anderson, 2017*; *Klein et al., 2014*). Both compartments are composed of a specialised stromal microenvironment dominated by thymic epithelial cells (TECs). Negative selection is facilitated by promiscuous gene expression (PGE) in TEC, especially so in medullary TEC (mTEC) that express the autoimmune regulator, AIRE (*Sansom et al., 2014*). This selection ultimately leads to a diverse but self-tolerant T cell receptor (TCR) repertoire.

Thymic size is already compromised in humans by the second year of life, decreases further during puberty, and continuously declines thereafter (*Kumar et al., 2018*; *Linton and Dorshkind, 2004*; *Palmer, 2013*). With this reduced tissue mass, cell numbers for both lymphoid and epithelial cell compartments decline. This is paralleled by an altered cellular organisation of the parenchyma, and the accumulation of fibrotic and fatty changes, culminating in the organ's transformation into adipose tissue (*Shanley et al., 2009*). Over ageing, the output of naive T cells is reduced and the peripheral lymphocyte pool displays a progressively altered TCR repertoire (*Egorov et al., 2018*; *Thome et al., 2016*). What remains unknown, however, is whether stromal cell states and subpopulations change during ageing and, if so, how these changes impact on thymic TCR selection.

To resolve the progression of thymic structural and functional decline, we studied TEC using single-cell transcriptomics across the first year of mouse life. We investigated how known and previously unrecognized TEC subpopulations contribute to senescence of the stromal scaffold and correspond to alterations of thymocyte selection and maturation. Our results reveal how the altered transcriptomes of mature TEC subtypes reflect the functional changes they are undergoing with advancing age. Unexpectedly, we discovered that the quiescence of TEC progenitors is a major factor underlying thymus involution. These findings have consequences for targeted thymic regeneration and the preservation of central immune tolerance.

## Results

### Thymus function is progressively compromised by age

Thymus morphological changes were evident by 4 weeks of age in female C57BL/6 mice, including cortical thinning and the coalescence of medullary islands (*Figure 1a*). These gross tissue changes coincided with changes in thymocyte and TEC cellularity (*Figure 1b,c* and *Figure 1—figure supplement 1*), as noted previously (*Gray et al., 2006*; *Manley, 2011*; *Ki et al., 2014*). Total thymic and TEC cellularity halved between 4 and 16 weeks of age (*Figure 1b,c* and *Figure 1—figure supplement 1*).

Given this sharp decline in TEC cellularity, we investigated whether the primary function of the thymus was compromised. Using flow cytometry, we profiled developing thymocytes undergoing negative selection (Helios+: Materials and methods; *Figure 1—figure supplement 2a*), a process that can be partitioned into four key stages: (1) double positive thymocytes (Wave 1a: Helios+PD-1+), (2) immature CD4+/CD8+ single positive (SP) thymocytes (Wave 1b: Helios+PD-1+), (3) semi-mature CD4+/CD8+ SP thymocytes (Wave 2a: Helios+) and, (4) mature CD4+/CD8+ SP thymocytes (Wave 2b: Helios+; *Figure 1d*, left panel) (*Daley et al., 2013*). Across these stages, the frequency of negatively selected thymocytes varied with age (*Figure 1d*, right panel). Specifically, negative selection of MHC class I restricted (i.e. CD8+ SP) thymocytes increased after the first week of life (Wave 1b) (*Figure 1d*, *Figure 1—figure supplement 2b,c*). In contrast, the proportions of both CD4+ and CD8+ semi-mature thymocytes undergoing negative selection in the medulla diminished with age (*Figure 1d*).

Impaired negative selection in the medulla undermines the production of a self-tolerant TCR repertoire. Using TCR-targeted bulk sequencing of the most mature CD4+ SP thymocytes (denoted 'M2') (*James et al., 2018*), we observed that 1-week-old mice exhibited shorter CDR3 lengths and a lower proportion of non-productive TCR α and β chain sequences than older mice (*Figure 1—figure supplement 2e,f*). V(D)J segment usage was altered by age (*Figure 1e,f*), which has the potential to reshape the antigen specificity repertoire of newly generated T cells. Approximately one-third of β chain V or J segments showed an age-dependent use (38% and 29%, respectively), illustrating the robustness of TCR V(D)J usage to thymic involution and the decline in thymocyte negative selection. Diversity of the TCR repertoire amongst the most mature thymocytes, however, increased significantly over age (*Figure 1g*), although not monotonically, along with the incorporation of more non-

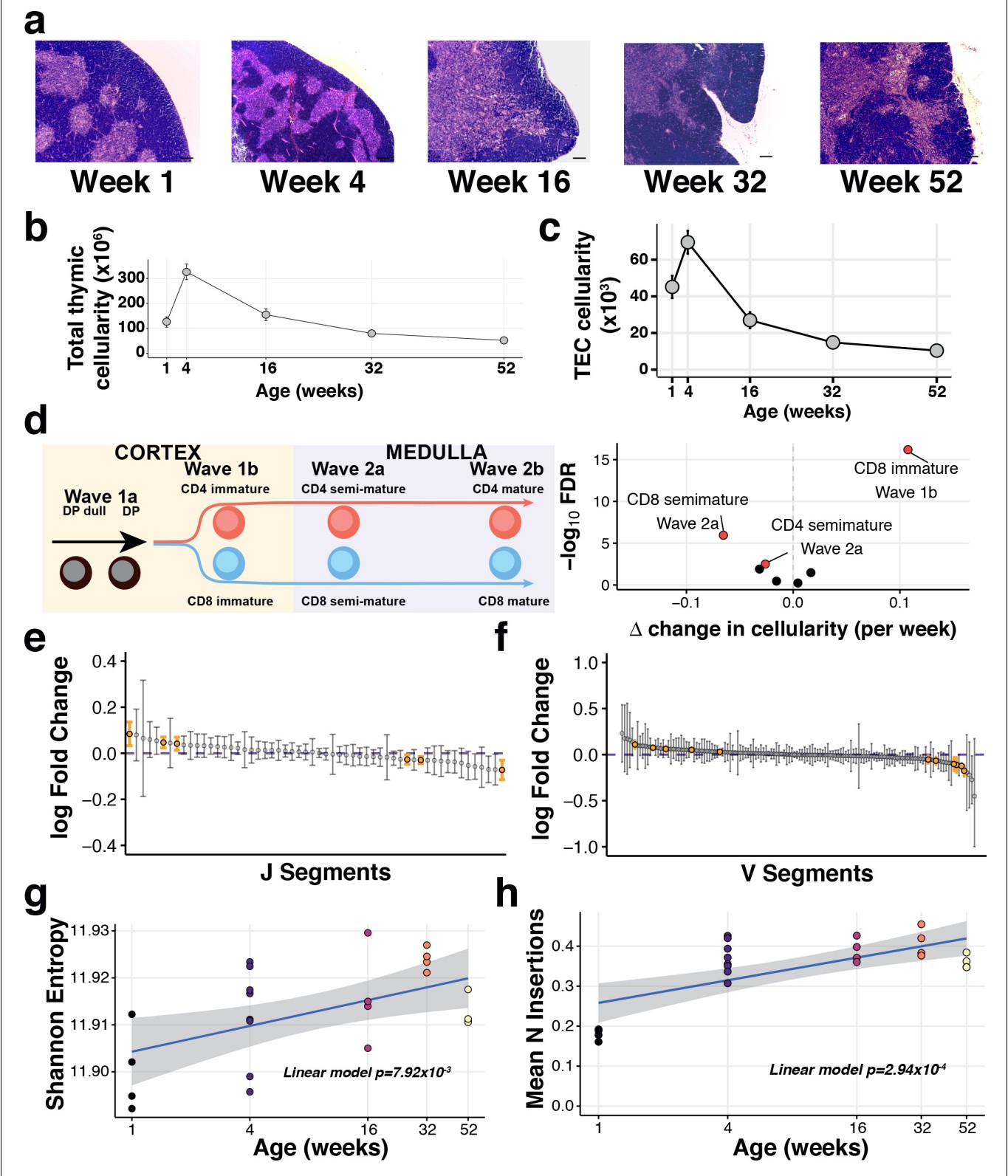

**Figure 1.** The decline of thymic cellularity and immune function with age. (**a**) Age-dependent changes in thymic architecture, as shown by representative H and E staining of thymic sections. Scale bars represent 150 μm. Medullary islands stain as light purple while cortical regions stain as dark purple. (**b**) Total and (**c**) TEC cellularity changes in the involuting mouse thymus. Error bars represent mean +/- standard error (five mice per age). (**d**) Thymocyte-negative selection declines with age: (Left) Schematic showing the progression of T cell development and Wave selection stages in the

*Figure 1 continued on next page*

Figure 1 continued

thymus that were investigated; (Right) Volcano plot showing the differential abundance of each of these thymocyte negative selection populations over age. Populations that are statistically significantly altered with age (FDR 1%) are labelled and highlighted in red. (e–f) The distribution of log-fold changes showing the alterations in individual TCR J segment (e) and V segment (f) usage with age. Log fold changes +/- 99% confidence intervals are plotted, with differentially abundant segments coloured in orange. (g) Mature thymocyte TCR repertoire diversity changes with age. The y-axis indicates the Shannon entropy of M2 thymocyte TCR CDR3 clonotypes at each age (n = 4–7 mice per time point), derived from TCR-sequencing of ~15,000 cells per sample. p-Value has been calculated from a linear model that regresses Shannon entropy on age. (h) The number of non-templated nucleotide insertions detected by TCR sequencing increases with age. The displayed p-value is from a linear model that regresses the mean number of inserted nucleotides on age.

The online version of this article includes the following figure supplement(s) for figure 1:

**Figure supplement 1.** Alterations in TEC composition throughout ageing.
**Figure supplement 2.** Changes in T-cell populations throughout ageing.
**Figure supplement 3.** T cell receptor repertoire simulations.

templated nucleotides (*Figure 1h*). The latter is inversely correlated with the post-puberty decline in the expression of thymocyte terminal deoxynucleotidyl transferase (*Cherrier et al., 2002*), suggesting an ageing-altered mechanism that is not intrinsic to the developing T cells. Taken together, these dynamic changes indicate that the principal immune functions of the thymus are progressively compromised with involution.

## Ageing remodels the thymic stromal epithelium

To determine whether the different TEC subpopulations were indiscriminately affected by ageing, we identified and analysed four major mouse TEC (CD45⁻EpCAM⁺) subpopulations at five postnatal ages using flow cytometry (*Supplementary file 1*-table 1) (*Gray et al., 2002*; *Wada et al., 2011*): cortical TEC (cTEC), immature mTEC (expressing low cell surface concentrations of MHCII, designated mTEC$^{lo}$), mature mTEC (mTEC$^{hi}$) and terminally differentiated mTEC (i.e. mTEC$^{lo}$ positive for desmoglein expression, Dsg3+ TEC) (*Figure 2a* and *Figure 2—figure supplement 1a*). Following index-sorting, SMART-Seq2 single-cell RNA-sequencing, and quality control (*Figure 2—figure supplement 1b–h*), we acquired 2327 single-cell transcriptomes, evenly distributed across the four cytometrically defined subpopulations and the five ages.

Our analysis revealed nine TEC subtypes (*Figure 2b,c*), thus providing a greater richness of epithelial states than previously appreciated (*Bornstein et al., 2018*; *Park et al., 2020*; *Figure 2d*, *Figure 2—figure supplement 2*) and a greater diversity than the four phenotypes cytometrically defined and selected in this study (*Figure 2—figure supplement 3a–b*). The individual subtypes, which are also identifiable from two independent data sets (*Figure 2d*, *Figure 2—figure supplement 2*; *Bornstein et al., 2018*; *Park et al., 2020*), were distinguished by the expression of marker genes (*Figure 2c* and *Supplementary file 1*-table 2). Among these were markers well established for different TEC populations (post-AIRE mTEC: *Krt80, Spink5*; Mature cTEC: *Prss16, Cxcl12*; mature mTEC: *Aire, Cd52*) and others that have been described more recently (Tuft-like mTEC: *Avil, Trpm5*) (*Bornstein et al., 2018*; *Miller et al., 2018*). Importantly, each TEC subtype, as defined by its single-cell transcriptome (*Figure 2b*), did not segregate exclusively with a single cytometrically defined TEC population (*Figure 2—figure supplement 3a*, *Supplementary file 1*-table 3). For example, a subtype that we termed Intertypical TEC (*Ccl21a, Krt5*; *Supplementary file 1*-table 3), and which was evident at all postnatal time-points, was composed of cells from each of the four cytometrically defined TEC subpopulations. Hereafter, for clarity, we refer to transcriptomically defined TEC clusters as subtypes and cytometrically specified TEC as subpopulations.

Four novel TEC subtypes were identified (*Supplementary file 1*-table 4): perinatal cTEC (marked by the expression of *Syngr1, Gper1*), Intertypical TEC (*Ccl21a, Krt5*) and two rare subtypes, termed neural TEC (nTEC: *Sod3, Dpt*) and structural TEC (sTEC, *Cd177, Car8*) based on their enrichment of neurotransmitter and extracellular matrix expression signatures (e.g. *Col1a1, Dcn, Fbn1*), respectively (*Figure 2—figure supplements 4–6*). Specifically, nTEC both lacked expression of *Rest* (RE1 silencing transcription factor), a transcriptional repressor that is typically expressed in all non-neuronal cells (*Nechiporuk et al., 2016*), and expressed genes silenced by REST (*Snap25, Chga, Syp*), markers shared in common with enteroendocrine cells of the small intestine (*Borges et al., 2010*; *Engelstoft et al., 2015*). Similarly, sTEC were marked by expression of numerous collagens and

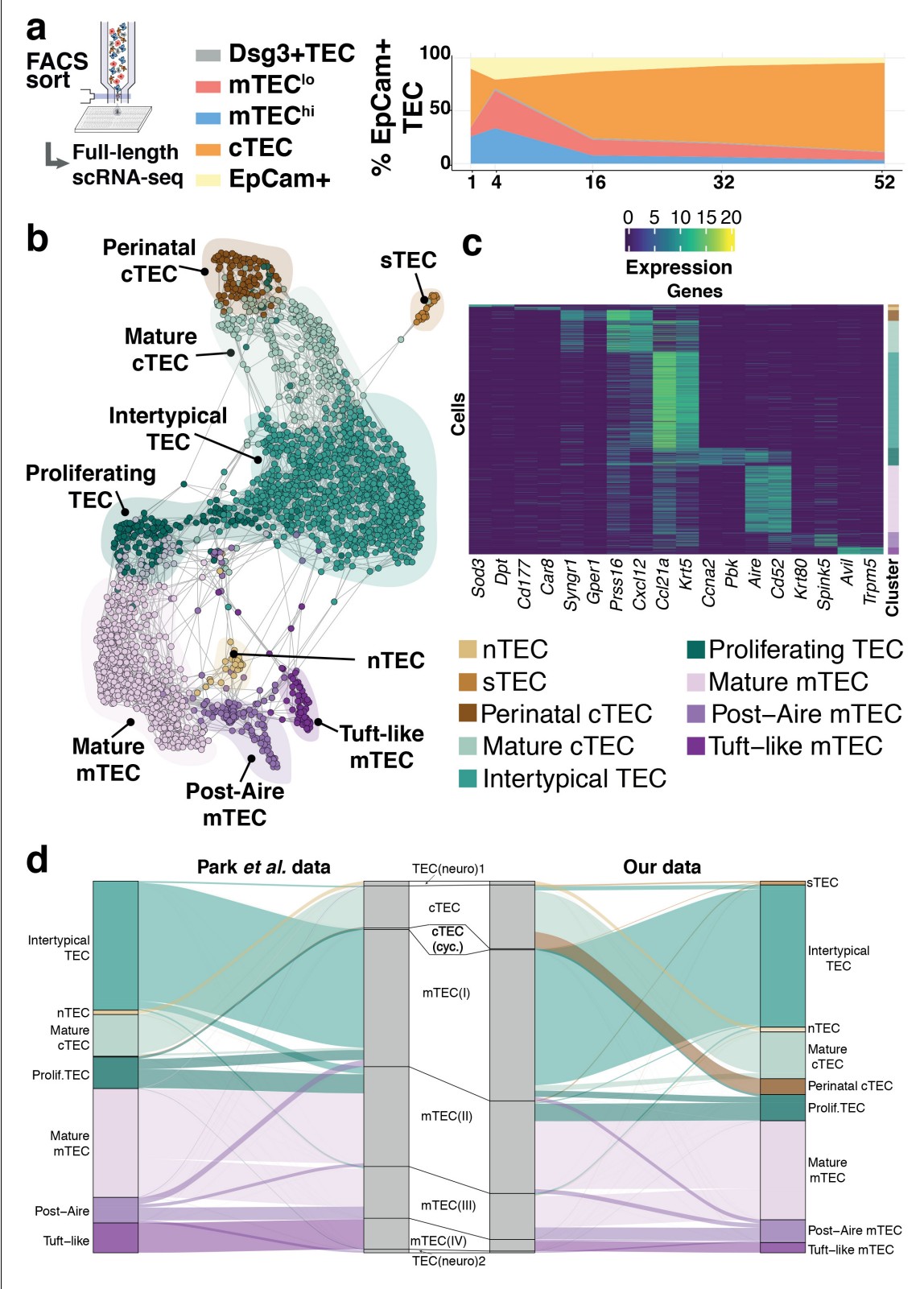

**Figure 2.** Thymic stromal composition remodelling during ageing. (a) A schematic showing the experimental design and FACS phenotypes of sorted cells for single-cell RNA-sequencing. Right panel shows cell composition fluctuations as a relative fraction of all EpCAM+ TEC with respect to the TEC subsets investigated. Remaining EpCAM+ cells not FACS-sorted are represented in the EpCAM+ population. (b) A SPRING-layout of the shared nearest-neighbour graph of single TEC, derived from scRNA-seq transcriptional profiles. Graph nodes represent single cells and edges represent

*Figure 2 continued on next page*

*Figure 2 continued*

shared k-nearest neighbours (k = 5). Cells are coloured by a clustering that joins highly connected networks of cells based on a random walk (Walktrap *Pons and Latapy, 2005*). Clusters are annotated based on comparisons to known TEC subsets and stereotypical expression profiles. (**c**) A heatmap of marker genes for TEC subtypes identified from single-cell transcriptome profiling annotated as in (**b**). (**d**) River plots illustrating the mapping between our TEC subtype labels and those from Park et al., using k-NN classifiers. Each line represents a single cell from the respective data set (postnatal and adult mice from Park et al.), where the central pillar shows the TEC labels from Park et al., and the outer pillars show the labels for our TEC subtypes; lines are coloured according to the TEC subtype labels in (**b**).

The online version of this article includes the following figure supplement(s) for figure 2:

**Figure supplement 1.** Experimental investigation of the ageing thymus.

**Figure supplement 2.** Comparison of *Bornstein et al., 2018*, *Park et al., 2020* single-cell transcriptomes to the TEC subtypes defined in this study.

**Figure supplement 3.** Relationship between classical FAC sorted subpopulations and transcriptionally defined single-cell subtypes.

**Figure supplement 4.** Expression of selected marker genes overlaid on SPRING plot.

**Figure supplement 5.** MSigDB pathways enriched for expression of marker genes for each single-cell subtype.

**Figure supplement 6.** Reactome pathways enriched for expression of marker genes for each single cell subtype.

**Figure supplement 7.** Consensus clustering of ageing single-cell TEC libraries.

proteoglycans indicating a possible role in extracellular matrix maintenance within the thymus. Perinatal cTEC were derived almost exclusively from the cytometric cTEC population and expressed $\beta-5t$ (encoded by *Psmb11*), which is both a component of the cortical thymoproteosome and a marker expressed by TEC progenitors (*Mayer et al., 2016*; *Ohigashi et al., 2013*). In addition to sharing many of the classical cTEC markers (*Figure 2c*; *Prss16*, *Cxcl12*, *Figure 2—figure supplement 4*), perinatal cTEC were characterised by a highly proliferative transcriptional signature (*Figure 2—figure supplement 6*). In contrast, Intertypical TEC were present in both cortical and medullary subpopulations, and expressed gene markers associated with a progenitor-like TEC$^{lo}$ phenotype (*Supplementary file 1*-table 4). Importantly, Intertypical TEC (including those derived from cortical subpopulations) displayed a restricted expression of $\beta-5t$ yet had transcripts for other canonical cTEC markers (e.g. *Cxcl12*, *Ctsl*). This illustrates that the use of conventional cTEC FAC-sorting strategies, (e.g. selection on Ly51 expression) captures a highly heterogeneous mixture of cells. Moreover, we found that Intertypical TEC expressed markers concordant with previously described TEC populations that localise to the corticomedullary junction (CMJ), and have been ascribed with progenitor-like functions (*Mayer et al., 2016*). These populations have variously been labelled as (*Supplementary file 1*-table 4): junctional TEC (jTEC; Pdpn, Ccl21a; *Onder et al., 2015*), TPAlo (Ccl21a; *Michel et al., 2017*) and Sca1+ (Ly6a, Plet1, Itga6; *Ulyanchenko et al., 2016*; *Lepletier et al., 2019*). Together, these results suggest: (i) that previous classifications annotated a heterogeneous population containing Intertypical TEC, and (ii) that Intertypical TEC might provide the progenitors of mature mTEC found at high density at the CMJ (*Onder et al., 2015*; *Michel et al., 2017*; *Ulyanchenko et al., 2016*; *Lepletier et al., 2019*; *Mayer et al., 2016*; *Supplementary file 1*-table 4).

To investigate how involution affected expression changes within each subtype as well as relative changes in the abundance of individual subtypes, we identified genes that changed expression in an age-dependent manner (*Figure 3a*, *Figure 3—figure supplement 1*) and modelled TEC subtype abundance as a function of age (*Figure 3b,c*). The cellular abundance of most TEC subtypes (6 of 9) varied significantly over age (*Figure 3b,c*; Materials and methods). For example, perinatal cTEC represented approximately one-third of all TEC at week 1 (*Figure 3b*) but contributed less than 1% three weeks later. Conversely, the proportion of mature cTEC and Intertypical TEC increased over time reaching ~30% and~60% of all TEC, respectively, by 1 year.

Gene expression signatures that are characteristic of ageing across diverse organs and species have been reported (*Benayoun et al., 2019*). Many of these signatures were also evident in the transcriptomes of individual ageing TEC subtypes (*Figure 3a*). For example, as they aged, mature mTEC genes involved in inflammatory signalling, apoptosis and increased KRAS signalling were up-regulated, whereas genes involved in cholesterol homeostasis and oxidative phosphorylation were downregulated (*Figure 3a*, left and right panels, respectively). In contrast, Intertypical TEC displayed an opposite pattern (*Figure 3a*, left panel, dark green bars): their ageing-related decrease in cytokine signalling pathways contrasted with the stronger inflammatory signature characteristic of senescent tissues, a.k.a. inflamm-ageing (*Franceschi et al., 2006*). In summary, mouse thymus involution is

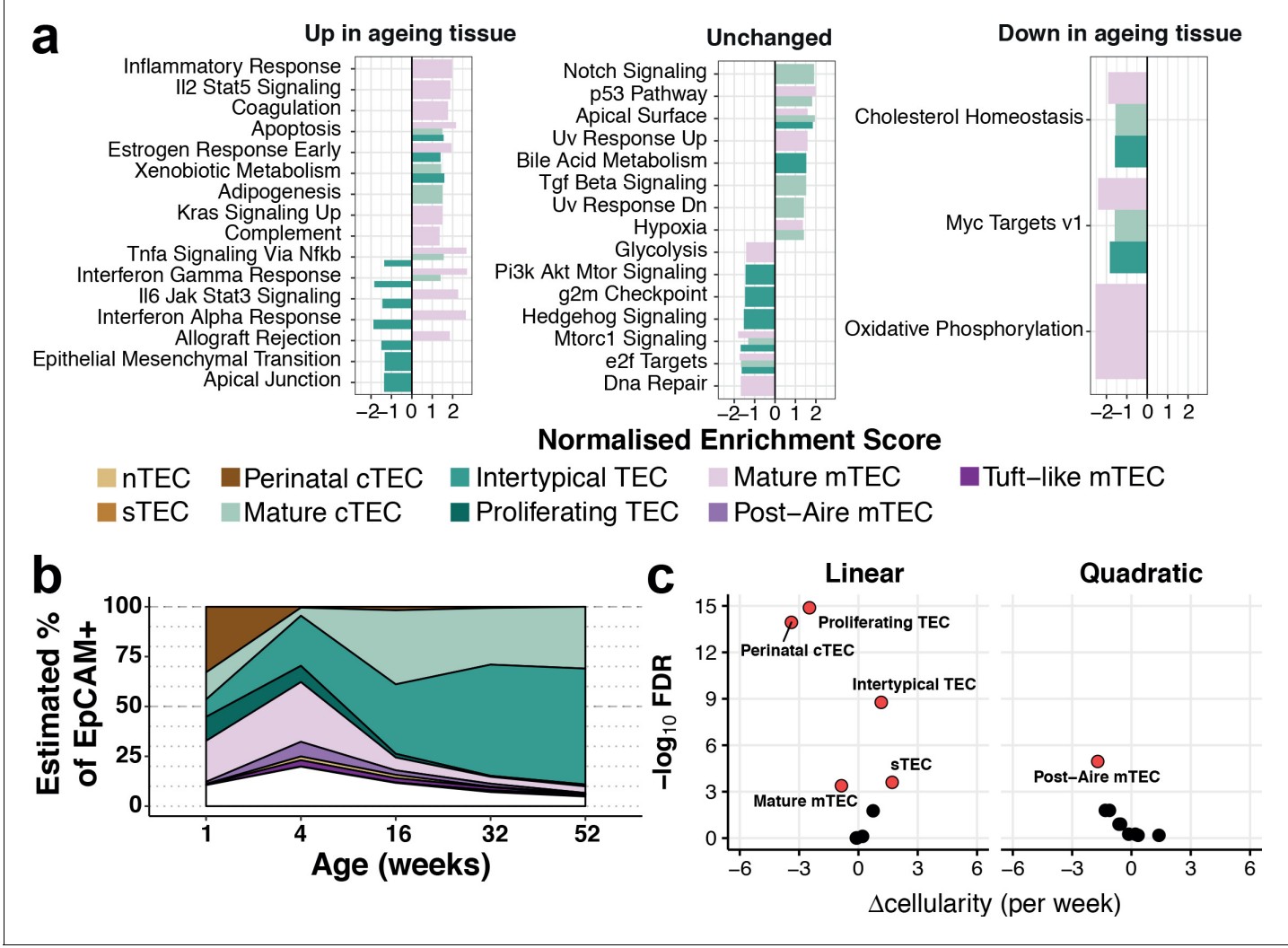

**Figure 3.** Thymic stromal remodelling during ageing. (a) Enrichment of MSigDB biological pathways with age in mature cTEC, intertypical TEC and mature mTEC, annotated as in (*Figure 2b*). Bars denote normalised enrichment score (NES) for significant pathways (FDR 5%), with enrichments coloured by cell type. Age-related alterations are shown in the context of pathways that are up-regulated (left), down-regulated (right) or do not change (middle) across multiple tissues and species (*Benayoun et al., 2019*). (b) A ribbon-plot demonstrating the compositional changes in TEC subtypes across ages, as an estimated fraction of all TEC (EpCAM+). Colours indicating each subtype are shown as in *Figure 2* above the plot with unsorted TEC indicated in white. (c) A volcano-plot of a negative binomial generalised linear model (GLM) showing linear (left) and quadratic (right) changes in cell cluster abundance as a function of age. X-axis denotes the change (Δ) in cellularity per week, and the Y-axis shows the -log₁₀false discovery rate (FDR). Subtypes with statistical evidence of abundance changes (FDR 1%) are labelled and shown as red points.

The online version of this article includes the following figure supplement(s) for figure 3:

**Figure supplement 1.** Differential expression of genes throughout ageing.

mirrored by alterations in both TEC subtype composition and transcriptional states. The transcriptional signature of inflamm-ageing was restricted to mature cTEC and mTEC (*Figure 3—figure supplement 1*) and an altered subtype frequency was most striking for Intertypical TEC and perinatal cTEC.

A principal function of mTEC is the promiscuous expression of genes encoding self-antigen (*Figure 4a*), which was also altered across age (*Figure 4b*). PGE is facilitated in part by the protein AIRE, which leads to the expression or enhanced expression of tissue restricted antigen-genes (TRAs) (*Sansom et al., 2014*). We did not observe an altered expression of genes reported to regulate PGE (*Figure 4c*: *Aire* and *Fezf2 Takaba et al., 2015*). However, PGE of AIRE-controlled genes was diminished at later ages, even when *Aire* transcripts persisted, suggesting a mechanism of

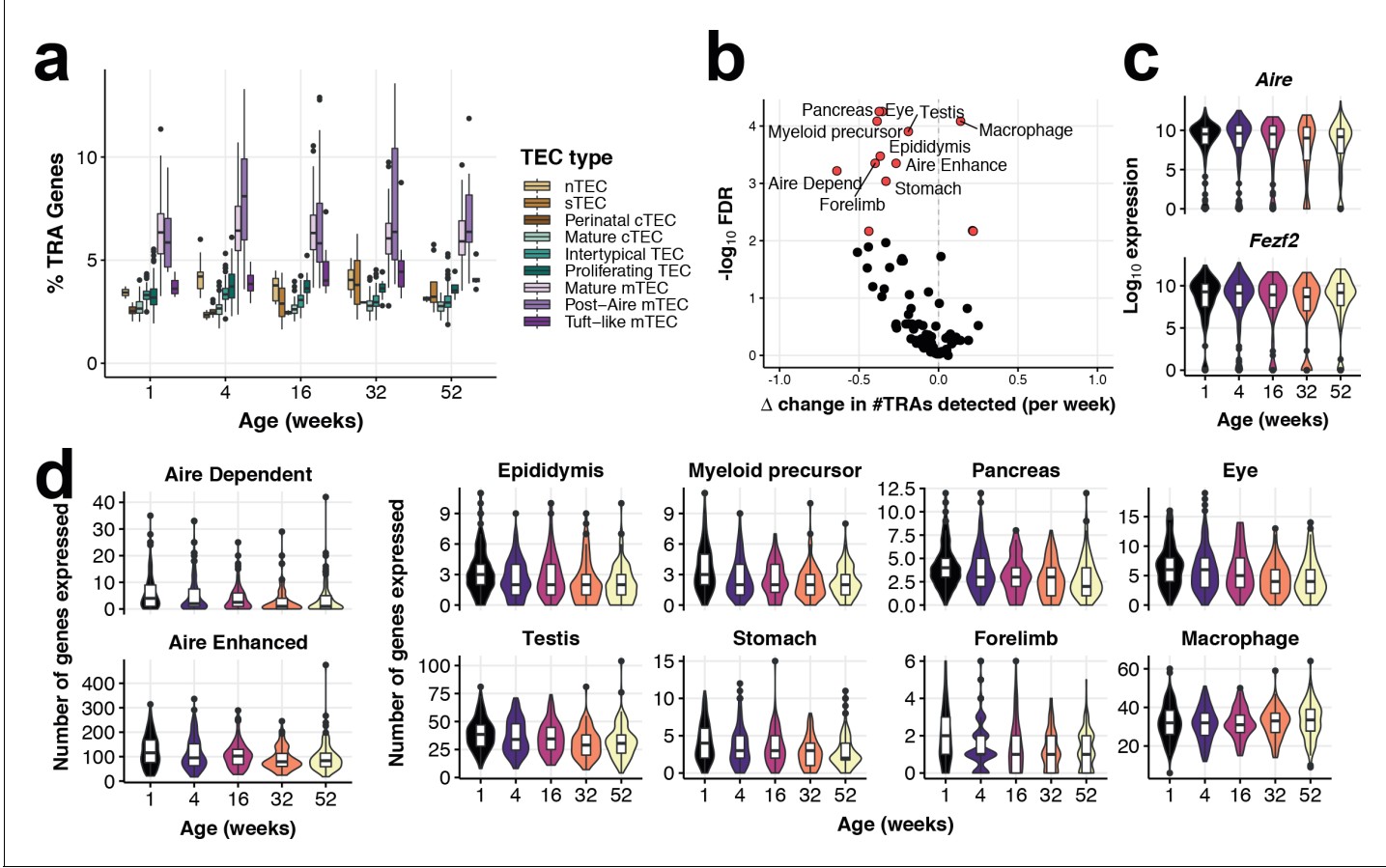

**Figure 4.** Tissue-restricted gene expression in mTEC. (a) TRA expression enriched in Mature mTEC and Post-Aire mTEC. Shown is the percentage of expressed genes that are classed as TRAs (see Materials and methods), for each TEC subtype across mouse ageing. (b) A volcano plot of differential TRA abundance testing, showing the consistent down-regulation of TRAs in mature mTEC. The X-axis denotes the change (Δ) in TRAs detected per week, and the Y-axis shows the -log10 false discovery rate (FDR). Tissue types or categories of promiscuously gene expression (PGE) with statistical evidence of abundance changes (FDR 1%) are labelled and shown as red points. (c) Violin plots of mTEC single-cell gene expression distributions across mouse ages of known promiscuous gene expression (PGE) regulators, *Aire* and *Fezf2*. (d) Violin plots of the number of mTEC-expressed tissue antigen genes in TRA groups that are differentially abundant over age, as in (b).

transcription that is also reliant on factors other than AIRE abundance (*Figure 4d*). Antigen transcripts restricted to particular tissues displayed striking reductions in age-related expression, including eye, pancreas and epididymis (*Figure 4d*). Notable exceptions to this general pattern of reduced TRA expression were macrophage-associated transcripts involved in inflammatory cytokine signalling, as noted above, whose expression increased over age (*Figure 3a*, *Figure 4b,d*). In summary, PGE in mature mTEC, and thus their capacity to represent 'Self', is increasingly compromised with age contributing to the thymic functional decline.

## Ageing compromises the differentiation of intertypical TEC into mature mTEC

Two lines of evidence indicated that Intertypical cells represent a TEC progenitor state. The first is that they exhibited a transcriptional signature that includes marker genes for both mature cTEC and mature mTEC (*Supplementary file 1*-tables 2,4). The second is that they express marker genes associated with previously described mTEC precursors (jTEC: *Onder et al., 2015*; TPA<sup>lo</sup>: *Michel et al., 2017*, Sca1+: *Ulyanchenko et al., 2016*, *Lepletier et al., 2019*; *Supplementary file 1*-table 4). Mature mTEC are derived from progenitor cells located at the CMJ which express β−5t (encoded by *Psmb11*) (*Mayer et al., 2016*; *Ohigashi et al., 2013*). To investigate experimentally whether

Intertypical and mature TEC share a common progenitor, we lineage traced the progeny of β−5t+ TEC using a triple transgenic mouse (denoted 3xtg$^{β5t}$) with a doxycycline-inducible fluorescent reporter, ZsGreen (ZsG), under the control of the *Psmb11* promoter (*Figure 5a*; *Mayer et al., 2016*). Forty-eight hours after doxycycline treatment, we isolated ZsG+ mTEC (Ly51-UEA1+CD86-; *Figure 5—figure supplement 1*) from a 1-week-old mouse and profiled the traced cells using SMART-Seq2 scRNA-sequencing before comparing them with our reference atlas (*Figure 5*). These cells were grouped into three clusters (*Figure 5b*). Based on their expression of marker genes (*Figure 5c*), and a random forest classification of ZsG+ single-cells into TEC subtypes (*Figure 5d* and *Figure 5—figure supplement 2*), these ZsG+ cells were predominantly classified as either mature mTEC or Intertypical TEC (*Figure 5d–e*). Moreover, by integrating single-cells of different TEC subtypes from 1 week old mice we clearly demonstrated that ZsG+ cells lie on a common differentiation trajectory spanning from Intertypical TEC (top left; *Figure 5e*) to the mTEC lineage (bottom left). Together, these results are consistent, firstly, with Intertypical TEC and mature mTEC being derived from a common β−5t+ progenitor, and secondly, with the notion that Intertypical TEC include cells that have the potential to differentiate into mTEC.

Our current and previous observations indicate that TEC differentiation is highly dynamic across the mouse life course (*Figure 3b,c*; *Mayer et al., 2016*). We evaluated the dynamics of TEC differentiation by pulse-chase lineage tracing using 1, 4 and 16 week old 3xtg$^{β5t}$ mice for the analysis of mTEC and cTEC compartments that had been treated with doxycycline either 2 or 28 days earlier (*Figure 6a* and *Figure 6—figure supplement 1*). The majority of cTEC in 1 week old mice were labelled after 2 days, yet this fraction declined considerably in treated 4-week-old mice, concordant with the loss of perinatal cTEC and an emergence of mature cTEC over this time interval (*Figure 3b, c*); a dynamic that tracked with the differential cellularity of cTEC over age (*Figure 3b*). We noted an accumulation of mTEC 28 days after labelling 1-week-old mice (*Figure 6a*), which declined for mice labelled at 4 weeks old, at the peak of mTEC cellularity, reflecting a high turnover of mTEC over this period (*Mayer et al., 2016*; *Dumont-Lagacé et al., 2014*). These dynamics remained largely unchanged in 16–20 week old mice, concordant with observations of a longer half-life of mTEC across this age range (*Dumont-Lagacé et al., 2014*). The accumulation of ZsG+ cells 2 days after doxycycline treatment in 16-week-old mice was not maintained 4 weeks later in either cTEC or mTEC, evident by an approximately twofold reduction in labelled TEC (*Figure 6a*). These results suggest a compositional shift in both TEC compartments influenced by changes in TEC half-life (*Dumont-Lagacé et al., 2014*), precursor-progeny relationships and/or representation of TEC subtypes. We reasoned that the expansion of Intertypical TEC during ageing, reflecting its diminished capacity to differentiate into mature mTEC, could contribute to these lineage-tracing dynamics. This was motivated by the fact that (1) Intertypical TEC represent a precursor of mTEC (*Figure 5*) and (2) ageing Intertypical TEC displayed evidence of progressive quiescence, illustrated by the down-regulation of Myc target genes (*Figure 3a*), and the expression of *Itga6* (CD49f; *Supplementary file 1*-table 2), a marker of quiescent radioresistant TEC (*Dumont-Lagacé et al., 2017*).

Therefore, to explore how the relationships among progenitor, Intertypical and mature mTEC change with age we took advantage again of the 3xtg$^{β5t}$ mice. Thymi were labelled at weeks 1, 4 and 16 and harvested 4 weeks later in triplicate, sorting ZsG+ and ZsG- cTEC and mTEC in equal numbers (*Figure 6b* and *Figure 6—figure supplements 1* and *2*). Droplet single-cell RNA-sequencing and graph-based clustering of single-cells revealed considerable heterogeneity across TEC. We annotated individual clusters into TEC sub-clusters that reflected our TEC subtypes (*Figure 6c* - labels); this captured the majority of our previously observed TEC phenotypes (*Figure 6—figure supplements 3–5*). Amongst these sub-clusters we observed, and subsequently discarded, a group of cells whose shared expression of marker genes for both TEC and thymocytes was suggestive of physically interacting cells (*Figure 6—figure supplement 5*). We discovered that most sorted ZsG+ cells were concordant with an Intertypical TEC phenotype across all ages and that their proportion increased with age in line with their relative accumulation in the ageing thymus (*Figure 6d*). We then evaluated the frequency of ZsG+ cells within each TEC subtype (*Figure 6e*). Concordant with our lineage-tracing results, fewer ZsG+ mature cTEC (~30%) were captured from mice labelled at 1 week compared to older animals (≥80%, *Figure 6e*), in parallel with the shift from perinatal to mature cTEC in adult mice. The relative proportion of ZsG+ cells increased modestly in mature mTEC populations reflecting their longer half-life with advancing mouse age (*Dumont-Lagacé et al.,*

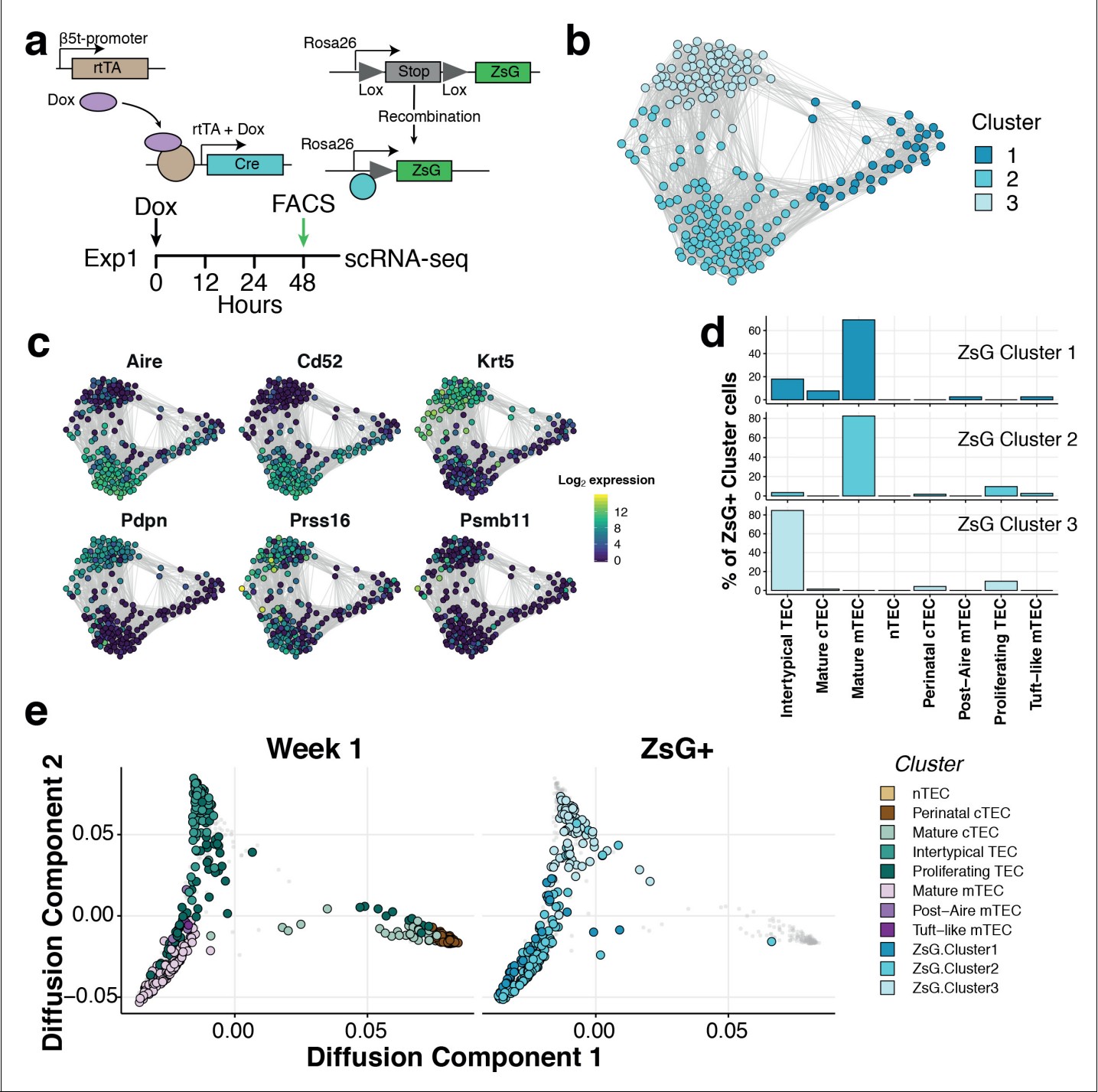

**Figure 5.** Intertypical TEC and medullary TEC are derived from a β−5t+ progenitor. (**a**) A schematic representing the transgenic Dox-inducible ZsGreen (ZsG) lineage tracing of β−5t-expressing mTEC precursors (top), and lineage tracing experiment in 1-week-old thymi (bottom). The green arrow denotes the interval post-Dox treatment. (**b**) A Fruchterman-Reingold layout of the shared nearest neighbor (SNN) graph of FACS-sorted ZsG+ mTEC from 1-week-old mice, 48 hr-post Dox treatment. Graph nodes represent cells coloured by a clustering of closely connected cells. (**c**) Expression levels of key medullary (*Aire*, *Cd52*), cortical (*Psmb11*, *Prss16*) and Intertypical TEC (*Krt5*, *Pdpn*) marker genes. (**d**) A β−5t-expressing precursor is the common origin of intertypical TEC and mature mTEC as shown by random forest classification of ZsG+ TEC. (**e**) A joint diffusion map between single TEC at week 1 (left panel), and ZsG+ TEC (right panel). Points represent single cells and are coloured by their assigned cluster as in *Figure 2* (week 1 TEC) or (**c**) (ZsG+ TEC).

The online version of this article includes the following figure supplement(s) for figure 5:

**Figure supplement 1.** Details of the ZsGreen labelling and sorting strategy.

*Figure 5 continued on next page*

*Figure 5 continued*

**Figure supplement 2.** Random Forest training using 1-week-old single TEC.

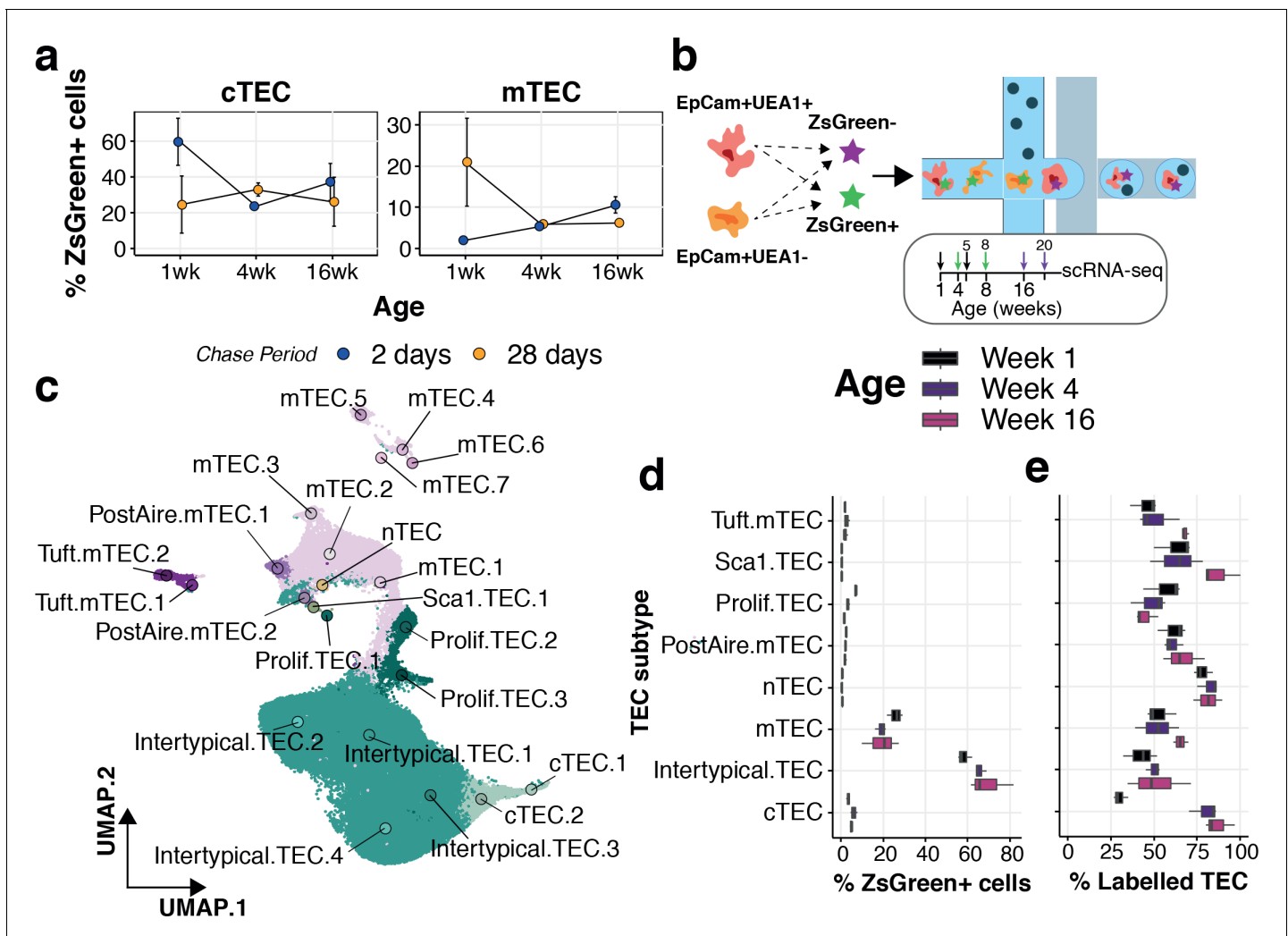

**Figure 6.** Lineage tracing and scRNA-seq reveal the dynamics of TEC precursor - progeny relationship across ages. (a) The mean percentage of cTEC (left) and mTEC (right) labelled either 2 days (blue points) or 28 days (orange points) after doxycycline treatment of 3xtg$^{\beta 5t}$ mice. Error bars are the mean +/- the standard error across five mice. (b) A schematic of the droplet single-cell RNA-sequencing of lineage traced cTEC and mTEC. Inset panel: schematic representation of ZsG lineage tracing of TEC across mouse ages. Paired colour arrows denote the time and age of doxycycline treatment and the subsequent collection of TEC (black: 1wk; green: 4wk; purple: 16wk). Numbers above the arrows represent the age of mice at the time of single-cell measurements. (c) A UMAP of all single TEC following quality control, coloured by an annotation into TEC subtypes according to the expression of TEC subtype marker genes. Filled circles are the average UMAP position of each TEC sub-cluster. (d) Box plots showing the distribution of ZsGreen+ sorted cells across TEC subtypes, coloured by mouse age at time of doxycycline treatment. (e) Box plots showing the percentage of cells in each TEC subtype that came from the ZsGreen+ fraction, coloured by mouse age at the time of doxycycline treatment.
The online version of this article includes the following figure supplement(s) for figure 6:

**Figure supplement 1.** Representative FACS plots depicting the gating strategy employed to determine the labelling efficiency of cTEC and mTEC of 1-week-old, 4-week-old and 16-week-old mice, 2 or 28 days after treatment with 0.004 mg (newborns) or 2 mg (adult mice) of doxycycline per mouse.
**Figure supplement 2.** Summary of multiplexed single-cell droplet RNA sequencing.
**Figure supplement 3.** Quality control of multiplexed single-cell droplet RNA sequencing.
**Figure supplement 4.** Droplet single-cell RNA-sequencing cluster annotation and comparison with TEC subtypes.
**Figure supplement 5.** Marker gene expression profiles across TEC clusters from β−5t lineage-traced single cells.
**Figure supplement 6.** UMAP visualisation of TEC sub-clusters across all single-cells from lineage-traced thymi.

*2014*). Notably, the exception to this pattern was a decline in ZsG+ proliferating TEC, concordant with the quiescence of this dividing mTEC state and the accumulation of Intertypical TEC.

To examine how TEC differentiation is altered with age, we used diffusion pseudotime analysis to connect single-cell transcriptomes across cortical and medullary lineages, starting from the Intertypical TEC compartment (*Figure 7a* and *Figure 7—figure supplements 1* and *2*). We observed and removed from this analysis Intertypical TEC sub-clusters that showed a differential bias for either cortical (Intertypical TEC 3) or medullary (Intertypical TEC 1 and 2) lineages based on marker gene expression (*Figure 6—figure supplement 5*). We identified a single cortical trajectory and a bifurcating medulla lineage originating from the Intertypical TEC cells (*Figure 7—figure supplements 1* and *2*). It has been established that cells can accumulate in meta-stable states along a differentiation trajectory and that the density of these cells is inversely related to the rate of their differentiation (*Haghverdi et al., 2016*). Consequently, we next followed the β−5t+ and β−5t- TEC states across age and examined how the density of cells changed in these meta-stable states (*Figure 7b,c* & *Figure 6—figure supplements 1* and *2*). While the cortical lineage demonstrated only a modest increase of ZsG+ cells at the most immature state by 16 weeks of age (*Figure 7—figure supplement 2*), for the medulla lineage we observed a considerable increase in ZsG+ cells over age in the earliest progenitor state (State 1 Poisson GLM $p<10^{-16}$), and a concordant decline in mature mTEC (State 3 Poisson GLM $p=7.25\times10^{-60}$), consistent with a partial block during differentiation (*Figure 7d*). These earliest regions of density in both lineages corresponded to the Intertypical TEC four sub-cluster (*Figure 7c* and *Figure 7—figure supplements 1* and *2*). Expression of characteristic genes along these trajectories demonstrated a peak of expression in the canonical TEPC maker *Ly6a* (SCA1)

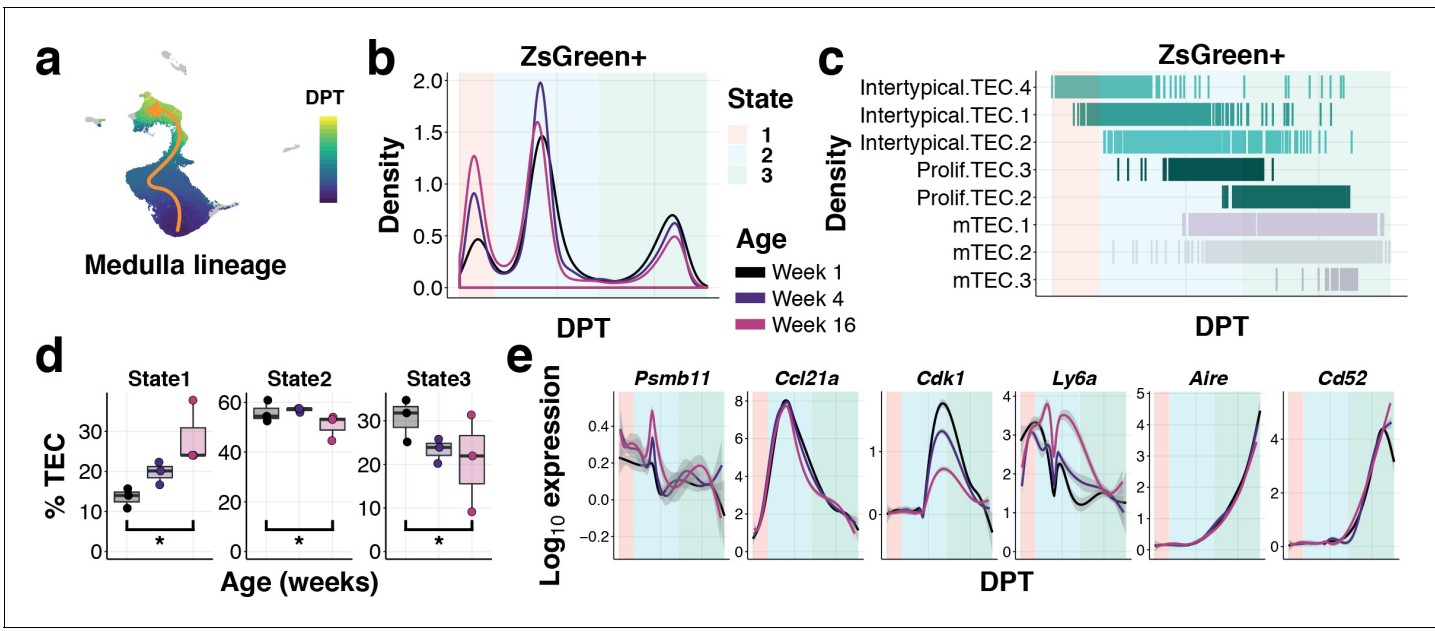

**Figure 7.** Ageing restricts the differentiation of intertypical TEC into mature mTEC. (**a**) UMAP as in (*Figure 6c*) coloured by diffusion pseudotime (DPT) distance in the medullary lineage (see *Figure 7—figure supplement 1* for details). (**b**) Changes in ZsGreen+ labelled TEC differentiation from the Intertypical TEC compartment as a function of age are shown as the density of cells ordered along pseudotime (DPT) in the medulla lineage, coloured by mouse age at time of doxycycline treatment. Three meta-stable TEC states defined by a Gaussian mixture model are highlighted in coloured rectangles. (**c**) A rug plot showing the TEC sub-clusters (y-axis) ordered along pseudotime (x-axis), coloured as in (*Figure 6c*). Shaded rectangles represent the span of the meta-stable states along DPT as in (**b**). (**d**) Boxplots showing the percentage of TEC coloured by age in mTEC differentiation states, as in (**b**). *denotes Poisson GLM p-values≤0.01 (see Methods). (**e**) Expression of TEC progenitor, Intertypical TEC and proliferating TEC markers across pseudotime. Each line shows the loess smooth expression coloured by age. Shaded rectangles represent the TEC meta-stable states shown in (**b**).

The online version of this article includes the following figure supplement(s) for figure 7:

**Figure supplement 1.** Pseudotime trajectory inference across medullary lineages.

**Figure supplement 2.** Pseudotime trajectory inference in the cortical lineage.

**Figure supplement 3.** Percentage of cells in each subtype with expression of key TEC genes.

within State 2 inline with a peak of *Ccl21a* expression (*Figure 7e*) prior to the up-regulation of classical cTEC (*Prss16, Cxcl12*; *Figure 7—figure supplement 2*) and mTEC genes (*Aire, Cd52*; *Figure 7e*). Notably, there was an early transient expression of *Psmb11* that occurred at the boundary of Intertypical TEC sub-clusters along the medulla lineage (*Figure 7e*).

In summary, by combining in vivo lineage tracing with single-cell transcriptome profiling, we have discovered that progenitor cells become increasingly blocked in Intertypical TEC states during ageing and that this reduced rate of maturation results in the decline of mTEC maintenance.

## Discussion

We have demonstrated how age re-models the thymic stromal scaffold and alters thymocyte composition thus impairing the core immunological function of the thymus. We find that as ageing progresses, fewer CD4+ and CD8+ SP thymocytes are negatively selected. The combination of overall reduction in mTEC cellularity and the decline in PGE of TRAs likely reduces the efficiency of antigen presentation thus impacting negative selection which, in turn, results in an increase in TCR repertoire diversity with age. With single-cell transcriptomics we identified nine TEC subtypes, four of which were previously undescribed (*Supplementary file 1*-table 4). These subtypes are largely concordant with TEC subpopulations identified recently (*Bornstein et al., 2018*; *Park et al., 2020*), but our investigation of mice at different ages has resolved the identity of some subpopulations that were amalgamated in previous studies (*Bornstein et al., 2018*; *Park et al., 2020*). Specifically, our data from 1-week-old mice provides a clear distinction between perinatal and mature cTEC. Similarly, our data from older mice demonstrates that Intertypical TEC (referred to as mTEC I in *Bornstein et al., 2018*; *Park et al., 2020*) are composed of both cytometrically sorted cTEC and mTEC$^{lo}$ populations. This study's refined categorisation of TEC subtypes highlights the insufficiency of previously established FACS-based and ontological TEC classifications, and should facilitate detailed investigations of their function using more specific markers (*Supplementary file 1*-table 2). By tracking TEC types and states across the murine life course, we have found that mature TEC subtypes exhibit age-altered gene expression profiles similar to those observed across many other tissues and species (*Benayoun et al., 2019*), as well as replicating previous observations of reduced cell proliferation and changes in global architecture (*Gray et al., 2006*; *Ki et al., 2014*). However, Intertypical TEC, a TEC subtype newly defined in this study, showed an opposing age-related pattern, with decreased expression in cytokine signalling pathways.

TEC progenitors have been described with distinctive molecular identities (e.g. β−5t expression) in the postnatal thymus where they dynamically expand and contribute to the mTEC scaffold (*Bleul et al., 2006*; *Ucar et al., 2014*; *Ulyanchenko et al., 2016*; *Wong et al., 2014*). During mouse development these TEC progenitors arise from the endoderm of the third pharyngeal pouch (mid-gestation) and subsequently develop into lineage-restricted cTEC and mTEC progenitors (*Baik et al., 2013*; *Gordon et al., 2004*; *Hamazaki et al., 2007*; *Ohigashi et al., 2013*; *Ripen et al., 2011*; *Rodewald et al., 2001*; *Rossi et al., 2006*; *Shakib et al., 2009*). The ability of β−5t+ TEC progenitors to expand and maintain the mTEC scaffold is progressively reduced in adolescent mice (*Mayer et al., 2016*). Using lineage tracing, we revealed how adult medulla-biased Intertypical TEC arise from a β−5t+ TEC progenitor population and likely represent a precursor to mature mTEC (*Figure 5d*). The sporadic expression of *Psmb11* amongst mTEC-biased Intertypical TEC could also explain the ZsGreen+ signal in these experiments. However, across multiple experiments (*Figure 5, 6*) we find that *Psmb11* is not universally expressed in Intertypical TEC, despite the fact that 75% of sorted ZsGreen+ cells from 8-20 week old mice are Intertypical TEC. Therefore, we conclude that the ZsGreen+ labelling of mTEC-biased Intertypical TEC most likely arises from a shared β−5t+ TEC progenitor state and that Intertypical TEC represent a previously missing precursor between β−5t+ progenitors and mature mTEC. Following this reasoning, we explored whether the potential restriction of mTEC differentiation was a consequence of the quiescence of these precursors. Using a combination of lineage-tracing, single-cell RNA-sequencing and pseudotime analysis we found that mTEC precursors in our data differentially accumulated in the ZsG+ fraction (*Figure 7b–d*). Moreover, we observed that an mTEC-biased subset of Intertypical TEC accumulated in the ZsG- fraction (*Figure 7—figure supplement 1e*). In this experiment ZsG- Intertypical cells were either older than 4 weeks and thus had not been labelled, but had already begun to quiesce, or were derived from a separate β−5t- progenitor. The latter inference is compatible with the conclusion that mTEC

differentiation from an Intertypical TEC precursor is restricted with age, but this model is unparsimonious as it requires that mTEC arise from two distinct progenitors, one that expresses *Psmb11* and another that does not, which converge on similar transcriptional programs. However, two lines of evidence suggest that this might indeed be the case: (1) ZsG cells are preferentially enriched in the medulla-biased Intertypical TEC two sub-cluster (*Figure 7—figure supplement 1*), which lacks *Psmb11* expression, unlike the other Intertypical TEC groups (*Figure 7—figure supplement 2* and *3*), (2) we observed two differentiation trajectories amongst the mTEC arising from the Intertypical TEC compartment (*Figure 7—figure supplement 1*); the latter lineage is preferentially enriched for ZsG- mature mTEC (*Figure 6—figure supplement 6*). In sum, these observations indicate that the age-related expansion of Intertypical TEC is a direct consequence of their constrained differentiation into mature mTEC. Finally, our results beg the question of what molecular mechanism leads to Intertypical TEC quiescence and restricted mTEC differentiation? A recent study (*Lepletier et al., 2019*) suggests that TEC progenitors are re-programmed by interactions between BMP, Activin A and follistatin. In our data, *Fst* (encoding follistatin), *Bmp4* and *Inhba* (encoding Activin A) are specifically expressed in the Intertypical TEC 4sub-cluster (*Figure 7—figure supplement 3*). If the model proposed by Lepletier et al. is correct, then TEC progenitors may be the architects of their own malfunction.

The re-modelling of TEC maturation and the progression of inflam-ageing both alter thymus function and result in increased TCR diversity with age (*Figure 1g*). Concomitantly, two processes - diminution of mature TEC cellularity and blockage of TEC maturation - contribute to reduced presentation of self-antigens to developing thymocytes and thus to a less efficient negative selection. This impairment is in keeping with features of age-related thymic involution: its overall reduction in naive T-cell output and an increased release of self-reactive T-cells (*Goronzy and Weyand, 2003*; *Palmer, 2013*). To compound these effects, the involuting thymus is also rapidly purged of its distinctive perinatal cTEC population (*Figure 3b*). The consequences of this are likely to be a further loss of antigen presenting cTEC and reduced support of thymocyte maturation. Taken together, we expect that these TEC changes impair the maintenance of central tolerance and could explain, at least in part, the increased incidence of autoimmunity with advancing age (*Candore et al., 1997*), in which the cumulative dysfunction of thymic central tolerance induction over time generates a slow drip feed of self-reactive T cells into the periphery. Moreover, CD4+ thymocytes have been shown to contribute to the maintenance of mature Aire+ mTEC (*Irla et al., 2008*), thus the overall decrease in CD4+ thymocyte cellularity with age (*Figure 1—figure supplement 1*) may also play a role in the loss of mature mTEC.

In summary, our results reveal how the altered transcriptomes of mature TEC subtypes reflect the functional changes they are undergoing with advancing age. An enhanced understanding of the molecular mechanisms that prevent progenitors from fully progressing towards mature mTEC should facilitate studies exploring therapeutic interventions that reverse thymic decline.

## Materials and methods

### Mice

Female C57BL/6 mice aged 1 week, 4 weeks, 16 weeks, 32 weeks, or 52 weeks were obtained from Jackson Laboratories, and rested for at least 1 week prior to analysis. 3xtg$^{β5t}$ mice [β−5t-rtTA::LC1-Cre::CAG-loxP-STOP-loxP-ZsGreen] mice were used for lineage-tracing experiments as previously described (*Mayer et al., 2016*). All mice were maintained under specific pathogen-free conditions and according to United Kingdom Home Office regulations or Swiss cantonal and federal regulations and permissions, depending where the mice were housed.

### Isolation of thymic epithelial cells and thymocytes

Thymic lobes were digested enzymatically using Liberase (Roche) and DNaseI (VWR). In order to enrich for TEC, thymic digests were subsequently depleted of CD45+ cells using a magnetic cell separator (AutoMACS, Miltenyi) before washing and preparation for flow cytometry. Thymocytes were isolated by physical disruption of thymic lobes using frosted microscope glass slides.

## Flow cytometry and cell sorting

Cells were stained at a concentration of 5–10 × 10^6 per 100 µl in FACS buffer (2% fetal calf serum in PBS or 5% bovine serum albumin in PBS). *Supplementary file 1*-table 5 provides details of antibody staining panels. Staining for cell surface markers was performed for 20 min at 4°C, except for CCR7 which was performed for 30 min at 37°C in a water bath prior to the addition of other cell surface stains. The FoxP3 Transcription Factor Staining Buffer Kit (eBioscience) was used according to the manufacturer's instructions in order to stain for intracellular antigens. Cell viability was assessed using DAPI staining or LIVE/DEAD Fixable Aqua Dead Cell Stain (Invitrogen). Samples were acquired and sorted using a FACS Aria III (BD Biosciences). For single-cell RNA-sequencing index sorting was used and cells were sorted into 384 well plates. Flow cytometry data was analysed using FlowJo V 10.5.3.

## TCR rearrangement simulations

Simulations of TCR germline rearrangements were used to estimate TCR-sequencing sample sizes. Sequential steps of α- and β-chain rearrangement were simulated to model β-selection and double negative thymocyte maturation prior to negative selection. We uniformly sampled V(D)J segments from the C57BL/6 TCR locus. For the TCR β-chain, variable (V) and diversity (D) segments were randomly selected from available sequences. For joining (J) segments, the TRBJ1 locus was selected on the first attempt, and TRBJ2 if a second attempt to rearrange was made. Consequently, the matching TRBC segment was selected based on the J segment that was chosen (either TRBC1 or TRBC2). For the concatenation of each segment pair, that is V-J, V-D, VD-J, randomly selected nucleotides were inserted between the adjoining segments, based on sampling from a Poisson distribution with $\lambda = 4$. The productivity of the rearranged β-chain was determined by the presence of a complete open reading frame (ORF) beginning with a canonical start codon ('ATG') in the selected V segment that spanned the full V(D)J and constant segments. In the event of a failed rearrangement, a second attempt was made using the TRBJ2 and TRBC2 segments. If either of these attempts produced a valid TCR β-chain, then under the principle of allelic exclusion the simulation proceeded to the α-chain rearrangement. However, if the second rearrangement failed to produce a valid TCR β-chain, the process was repeated for the second allele.

For the TCR α-chain, variable (V) and joining (J) regions were randomly selected from the available TCRA sequences. Following the same principle as above, if the simulated rearrangement failed to generate a valid TCR with a complete ORF spanning the V segment to the constant region then the simulation switched to the second allele. A successful TCR germline was recorded only in the event of both valid α- and β-chains. The complete simulation resulted in a valid α-chain in 40.2% of simulations, and a valid β-chain in 63.1% of simulations.

To calculate sample sizes for our TCR-sequencing experiments we simulated 1 million 'thymocytes', and sub-sampled 10, 100, 500, 1000, 5000, 10,000, 20,000, 50,000 and 100,000 cells, defined by a productive pair of TCR chains. To simulate replicates we ran these simulations with 10 different random initiations. To establish the required sample sizes, we calculated the proportions of V(D)J segment frequencies for α- and β-chains. Additionally, we calculated the TCR diversity at each sample size using the Shannon entropy across α- and β-chain CDR3 clonotypes, defined by the unique amino acid sequence. Results of simulations are shown in *Figure 1—figure supplement 3*.

## TCR sequencing

15,000 M2 thymocytes (TCRb^hi, CCR7+, MHCI+, CD69-, CD8-, CD4+, CD25-) were sorted and RNA extracted using the Qiagen RNeasy Micro kit. 10 ng of RNA was used to prepare bulk TCR-seq libraries using the SMARTer Mouse TCR a/b Profiling Kit (Takara) according to instructions. Libraries were sequenced on a MiSeq (300 base paired-end reads). Reads were trimmed using Trimmomatic, down-sampled to the smallest library size and aligned using MiXCR (version 3.0).

## Haematoxylin and eosin (H and E) staining of thymic sections

Thymic lobes were harvested and cleaned under a dissecting microscope before being fixed in 10% Formalin (Sigma) for 12–36 hr, depending on size, and dehydrated in ethanol. After fixation, the tissues were embedded in paraffin using an automated system (Tissue-Tek Embedding Centre, Sakura) and sectioned to a thickness of 8 µm. H and E staining was performed using an automated slide

stainer (Tissue-Tek DRS 2000, Sakura) and slides were visualised under a light microscope DM750 (Leica).

## Plate-based single-cell RNA-sequencing

### Lysis plates

Single thymic epithelial cells were index FACS-sorted into 384-well lysis plates. Lysis plates were created by dispensing 0.4 µl lysis buffer (0.5 U Recombinant RNase Inhibitor (Takara Bio, 2313B), 0.0625% Triton X-100 (Sigma, 93443–100 ML), 3.125 mM dNTP mix (Thermo Fisher, R0193), 3.125 µM Oligo-dT 30 VN (IDT, 5'AAGCAGTGGTATCAACGCAGAGTACT 30 VN-3') and 1:600,000 ERCC RNA spike-in mix (Thermo Fisher, 4456740) into 384-well hard-shell PCR plates (Biorad HSP3901) using a Tempest liquid handler (Formulatrix). All plates were then spun down for 1 min at 3220 g and snap frozen on dry ice. Plates were stored at −80°C until used for sorting. cDNA synthesis and library preparation. cDNA synthesis was performed using the Smart-seq2 protocol (*Picelli et al., 2014*). Briefly, 384-well plates containing single-cell lysates were thawed on ice followed by first strand synthesis. 0.6 µl of reaction mix 16.7 U/µl SMARTScribe TM Reverse Transcriptase (Takara Bio, 639538), 1.67 U/µl Recombinant RNase Inhibitor (Takara Bio, 2313B), 1.67X First-Strand Buffer (Takara Bio, 639538), 1.67 µM TSO (Exiqon, 5'-AAGCAGTGGTATCAACGCAGACTACATrGrG+G-3'), 8.33 mM DTT (Bioworld, 40420001–1), 1.67 M Betaine (Sigma, B0300-5VL), and 10 mM MgCl 2 (Sigma, M1028−10 × 1 ML)) were added to each well using a Tempest liquid handler. Bulk wells received twice the amount of RT mix (1.2 µl). Reverse transcription was carried out by incubating wells on a ProFlex 2 × 384 thermal-cycler (Thermo Fisher) at 42°C for 90 min and stopped by heating at 70°C for 5 min. Subsequently, 1.6 µl of PCR mix (1.67X KAPA HiFi HotStart ReadyMix (Kapa Biosystems, KK2602), 0.17 µM IS PCR primer (IDT, 5'-AAGCAGTGGTATCAACGCAGAGT-3'), and 0.038 U/µl Lambda Exonuclease (NEB, M0262L)) was added to each well with a Tempest liquid handler (Formulatrix). Bulk wells received twice the amount of PCR mix (3.2 µl). Second strand synthesis was performed on a ProFlex 2 × 384 thermal-cycler using the following program: 1. 37°C for 30 min, 2. 95°C for 3 min, 3. 23 cycles of 98°C for 20 s, 67°C for 15 s, and 72°C for 4 min, and 4. 72°C for 5 min. The amplified product was diluted with a ratio of 1 part cDNA to 9 parts 10 mM Tris-HCl (Thermo Fisher, 15568025), and concentrations were measured with a dye-fluorescence assay (Quant-iT dsDNA High Sensitivity kit; Thermo Fisher, Q33120) on a SpectraMax i3x microplate reader (Molecular Devices). These wells were reformatted to a new 384-well plate at a concentration of 0.3 ng/µl and a final volume of 0.4 µl using an Echo 550 acoustic liquid dispenser (Labcyte). If the cell concentration was below 0.3 ng/µl, 0.4 µl of sample was transferred. Illumina sequencing libraries were prepared using the Nextera XT Library Sample Preparation kit (Illumina, FC-131–1096) (*Darmanis et al., 2017*; *Tabula Muris Consortium et al., 2018*). Each well was mixed with 0.8 µl Nextera tagmentation DNA buffer (Illumina) and 0.4 µl Tn5 enzyme (Illumina), then tagmented at 55°C for 10 min. The reaction was stopped by adding 0.4 µl 'Neutralize Tagment Buffer' (Illumina) and spinning at room temperature in a centrifuge at 3220 X g for 5 min. Indexing PCR reactions were performed by adding 0.4 µl of 5 µM i5 indexing primer, 0.4 µl of 5 µM i7 indexing primer, and 1.2 µl of Nextera NPM mix (Illumina). PCR amplification was carried out on a ProFlex 2 × 384 thermal cycler using the following program: 1. 72°C for 3 min, 2. 95°C for 30 s, 3. 12 cycles of 95°C for 10 s, 55°C for 30 s, and 72°C for 1 min, and 4. 72°C for 5 min.

## Library pooling, quality control, and sequencing

Following library preparation, wells of each library plate were pooled using a Mosquito liquid handler (TTP Labtech). Row A of the thymus plates, which contained bulk cells, was pooled separately. Pooling was followed by two purifications using 0.7x AMPure beads (Fisher, A63881). Library quality was assessed using capillary electrophoresis on a Fragment Analyzer (AATI), and libraries were quantified by qPCR (Kapa Biosystems, KK4923) on a CFX96 Touch Real-Time PCR Detection System (Biorad). Plate pools were normalised to 2 nM and sequenced on the NovaSeq 6000 Sequencing System (Illumina) using 2 × 100 bp paired-end reads with an S4 300 cycle kit (Illumina, 20012866). Row A thymus pools were normalised to 2 nM and sequenced separately on the NextSeq 500 Sequencing System (Illumina) using 2 × 75 bp paired-end reads with a High Output 150 cycle kit (Illumina, FC-404–2002).

## Single-cell RNA-sequencing processing, quality control and normalisation

Paired-end reads were trimmed to a minimum length of 75nt using trimmomatic with a 4nt sliding window with a quality threshold of 15. Leading and trailing sequences were removed with a base quality score <3 (*Bolger et al., 2014*). Contaminating adaptors were removed from reads with a single seed mismatch, a palindrome clip threshold of 30 and a simple clip threshold of 10. Trimmed and proper-paired reads were aligned to mm10 concatenated with the ERCC92 FASTA sequences (Thermo Fisher Scientific) using STAR v2.5.3a (*Dobin et al., 2013*) and a splice-junction database constructed from the mm10 Ensembl v95 annotation with a 99nt overhang. Paired-end reads were aligned with the parameters: `-outSAMtype` BAM SortedByCoordinate `-outSAMattributes` All `-outSAMunmapped` Within KeepPairs; all other parameters used default values. Following alignment each single-cell BAM file was positionally de-duplicated using PicardTools *MarkDuplicates* with parameters: *REMOVE_DUPLICATES = true, DUPLICATE_SCORING_STRATEGY = TOTAL_MAPPED_REFERENCE_LENGTH* [http://broadinstitute.github.io/picard].

De-duplicated single-cell transcriptomes were quantified against exon sequences of the mm10 Ensembl v95 using featureCounts (*Liao et al., 2014*). Poor-quality single-cell transcriptomes were removed based on several criteria: contribution of ERCC92 to total transcriptome >40%, sequencing depth $<1 \times 10^5$ paired-reads and sparsity (% zeros)>97%. From this initial round of quality control 2780 cells were retained for normalisation and downstream analyses. Deconvolution-estimated size factors were used to normalise for sequencing depth across single cells, prior to a log10 transformation with the addition of a pseudocount (+1), implemented in *scran* (*Lun et al., 2016*).

## Single-cell clustering and visualisation

TEC from all ages and sort-types were clustered together using a graph-based algorithm that joins highly connected networks of TEC based on the similarity of their expression profile. To enhance the differences in the expression profile of individual TEC libraries, we first applied a text frequency-inverse document frequency (TF-IDF) transform (*Manning et al., 2008*) to the gene-by-cell expression matrix. This transform enhances the signal from rarely expressed genes (of particular importance would be those that are promiscuously expressed in TEC), while also lessening the contribution from widely expressed genes. The transformed matrix represents the product of the gene-frequency and the inverse-cell-frequency. To compute this transformed matrix, we first assigned the gene-frequency matrix as the log2 of normalised gene-by-cell expression matrix ($G_f = \log_2 (C)$; C is the normalised count matrix). Next, we computed the inverse-cell-frequency as the inverse frequency of detection of each gene ($ICF_x = \log10 (N / (1+E_x))$; N is the number of cells, $E_x$ is the number of cells expressing gene *X*). Finally, the product of the gene-frequency matrix and inverse-cell-frequency was computed ($GF\_ICF = G_f * ICF$). The highly variable genes from this transformed matrix were used to compute a shared nearest neighbor (SNN) graph (k = 10), and the clusters were identified using a random walk (Walktrap *Pons and Latapy, 2005*) of the SNN graph. To assess the robustness of the clusters, we also clustered cells without the TF-IDF transform and using a series of alternate parameters. We computed a consensus matrix to determine how often the identified TEC subtypes co-clustered. We found that the identified TEC sub-types were robustly co-clustered regardless of the parameters of the clustering that was applied (*Figure 2—figure supplement 7*). Visualisation of the connected graph was computed using the SPRING algorithm to generate a force-directed layout of the K-nearest-neighbor graph (k = 5) (*Weinreb et al., 2018*).

## TEC nomenclature comparison

The MARS-seq counts matrix from *Bornstein et al., 2018* were downloaded from Gene Expression Omnibus (GSE103967), subset to experiments from wild-type mice, and normalised using deconvolution size factors (*Lun et al., 2016*), after removing cells with low sequencing coverage (<1000 UMIs, sparsity ≥98%). The counts matrix from Park et al was downloaded from [doi: 10.5281/zenodo.3711134] and subset to the cells that passed QC and normalised using deconvolution size factors (*Lun et al., 2016*).

To map single-cells across datasets we constructed kNN classifiers (k = 5) implemented in the R package *FNN*, trained on the ageing and Park et al single-cell data. We first took the set of commonly expressed genes between all three studies, and performed a per-cell cosine normalisation on

each data set prior to batch-correction using mutual nearest neighbours (*Haghverdi et al., 2018*). These data were used as input to classify each single cell using TEC subtype annotations from either our study or from Park et al (*Figure 2—figure supplement 2*).

## Lineage-tracing experiments

### Short-term lineage-tracing experiment

One-week old 3xtg$^{β5t}$ mice (N = 5 per replicate and per age group) were treated with a single i.p. injection of 0.3 mg of Doxycycline (Sigma) diluted in Hank's Balanced Salt Solution (Life Technologies). Forty-eight hours later, thymi were enzymatically digested with Liberase (Roche) and DNase (Sigma) and enriched for EpCAM+ cells using a magnetic cell separator (AutoMACS, Miltenyi) as previously described. Resulting single-cell suspensions were stained with fluorescently labelled antibodies against CD45, CD326 (EpCAM), MHCII, Ly51 and CD86, as well as with the UEA1 lectin. Single CD45-EpCAM+MHCII+Ly51-UEA1+CD86-ZsGreen+ cells were sorted directly into 386-well plates in order to enrich for TEC recently committed to the medullary lineage and to reduce the frequency of mature mTEC labelled with ZsGreen as a result of promiscuous gene expression. 3xtg$^{β5t}$ mice were developed from: Psmb11$^{tm2.1(rtTA)Yout}$(MGI:5911440), B6;C-Tg(tetO-cre)LC1Bjd/BjdCnrm (RRID:IMSR_EM:00753), and B6.Cg-Gt(ROSA)$^{26Sortm6(CAG-ZsGreen1)Hze}$/J (RRID:IMSR_JAX:007906) as shown in *Figure 5a* and as previously described (*Mayer et al., 2016*).

### Long-term lineage-tracing experiment

One-week old 3xtg$^{β5t}$ mice were treated with a single i.p. injection of 0.004 mg of Doxycycline (Sigma) diluted in Hank's Balanced Salt Solution (Life Technologies), whereas older mice (4-week and 16-week old) were treated with two i.p. injections of Doxycycline (2 mg, each) on 2 consecutive days during which they were also exposed to drinking water supplemented with the drug (2 mg/mL in sucrose (5% w/v)). Four weeks later, single thymic epithelial cell suspensions were obtained by enzymatic digestion using Liberase (Roche), Papain (Sigma) and DNase (Sigma) in PBS as described in *Kim and Serwold, 2019*; *Mayer et al., 2016*. Prior to FAC-sorting, TEC were enriched for EpCAM-positivity using a magnetic cell separator (AutoMACS, Miltenyi), as described above. Enriched cells were then stained for the indicated cell surface antigens (*Supplementary file 1*-table 5) in conjunction with TotalSeq-A oligonucleotide-conjugated antibodies (BioLegend) to allow for barcoding and pooling of different TEC subpopulations. Subsequently, the labelled cells were sorted into the following four subpopulations: ZsGreen+ cTEC, ZsGreen- cTEC, ZsGreen+ mTEC, and ZsGreen- mTEC (*Figure 6—figure supplement 2a*). After sorting, the cell viability and concentration of each of the cell samples collected were measured using a Nexcelom Bioscience Cellometer K2 Fluorescent Viability Cell Counter (Nexcelom Bioscience).

## Droplet-based single-cell RNA sequencing

### Droplet-based single-cell RNA-sequencing

Equal cell numbers were pooled from each of the samples, and a total of 30000 cells were loaded per well onto a Chromium Single Cell B Chip (10X Genomics) coupled with the Chromium Single Cell 3' GEM, Library and Gel Bead Kit v3 and Chromium i7 Multiplex Kit (10X Genomics) for library preparation, according to the manufacturer's instructions. In short, the cell suspension was mixed with the GEM Retrotranscription Master Mix and loaded onto well number one on the Chromium Chip B (10x Genomics). Wells 2 and 3 were loaded with the appropriate volumes of gel beads and partitioning oil, respectively, after which the Chromium Controller (10X Genomics) was used to generate nanoliter-scale Gel Beads-in-emulsion (GEMs) containing the single cells to be analysed. The fact that cell samples containing six different hashtag antibodies were pooled together allowed us to overload the 10X wells with 30000 cells per well, aiming for a recovery of approximately 12000 single cells (40%) per well. This also allowed us to overcome the resulting increase in doublet rate by subsequently eliminating from further analysis any cell barcode containing more than one single hashtag sequence. Incubation of the GEM suspension resulted in the simultaneous production of barcoded full-length cDNA from poly-adenylated mRNA as well as barcoded DNA from the cell surface protein-bound TotalSeqA antibodies inside each individual GEM. Fragmentation of the GEMs allowed for the recovery and clean-up of the pooled fractions using silane magnetic beads. Recovered DNA was then amplified, and cDNA products were separated from the Antibody-Derived Tags

(ADT) and Hashtag oligonucleotides (HTO) by size selection. The amplified full-length cDNA generated from polyadenylated mRNA were fragmented enzymatically and size selection was used to optimise amplicon size for the generation of 3' libraries. Library construction was achieved by adding P5, P7, a sample index, and TruSeq Read 2 (read two primer sequence) via End Repair, A-tailing, Adaptor Ligation, and PCR. Separately, ADT and HTO library generation was achieved through the addition of P5, P7, a sample index, and TruSeq Read 2 (read two primer sequence) by PCR. Sequences of the primers designed for this purpose can be found in *Supplementary file 1*-tables 6 and 7.

## Library pooling, quality control, and sequencing

Library quality was assessed using capillary electrophoresis on a Fragment Analyzer (AATI). The concentration of each library was measured using a Qubit dsDNA HS Assay Kit (ThermoFisher Scientific), and this information was then used to dilute each library to a 2 nM final concentration. Finally, the different libraries corresponding to each sample set were pooled as follows: 85% cDNA + 10% ADT + 5% HTO, after which pooled libraries were sequenced on an Illumina NovaSeq 6000 using the NovaSeq 6000 S2 Reagent Kit (100 cycles) (Illumina).

## Droplet-based single-cell RNA sequencing processing, de-multiplexing and quality control

Multiplexed 10X scRNA-seq libraries were aligned, deduplicated and quantified using Cellranger v3.1.0. Gene expression matrices of genes versus cells were generated separately for each sample (i.e. each 10X Chromium chip well), as well as those for hashtag oligo (HTO) and antibody (ADT) libraries. Cells were called using emptyDrops, with a background UMI threshold of 100 (*Lun et al., 2019*). Experimental samples, that is replicates and ZsGreen-fractions, were demultiplexed using the assigned HTO for the respective sample (*Stoeckius et al., 2018*). Specifically, within each sample, the HTO fragment counts were normalised across cell barcodes for all relevant HTOs using counts per million (CPM). These CPMs were used to cluster cell barcodes using k-means with the expected number of singlet clusters, that is unique HTOs in the respective sample. To estimate a background null distribution for each HTO within a sample, we then selected the k-means partition with the highest average CPM for the HTO and excluded these cells, along with the top 0.5% of cells with the highest counts for the respective HTO. We then fitted a negative binomial distribution to the HTO counts for the remaining cells to estimate a threshold ($q$) at the 99th quantile. All cell barcodes with counts $\geq q$ were assigned this HTO. This procedure was repeated for each HTO within a sample. Cell barcodes that were assigned to a single HTO were called as 'Singlets', whilst cell barcodes assigned to >1 HTO were called as 'Multiplets'. Finally, cell barcodes with insufficient coverage across HTOs were called as 'Dropouts' (*Figure 6—figure supplement 2b*). Only 'Singlets' were retained for normalisation and downstream analyses.

Poor-quality cells barcodes were removed based on high mitochondrial content, defined within each sample as twice the median absolute deviation from the median mitochondrial fraction. Cell barcodes with low coverage (<1000 UMIs detected) were also removed prior to normalisation. Finally, deconvolution-estimated size factors were calculated to normalise across single cells, then log10 transformed with a pseudocount (+1), as implemented in *scran* (*Lun et al., 2016*).

## Droplet single-cell RNA sequencing clustering and annotation

Highly variable genes (HVGs) were defined across droplet single cells based on the estimated fit across cells between the mean log normalised counts and variance, at an FDR of $1 \times 10^{-7}$ (*Brennecke et al., 2013*). The first 20 principal components (PCs) across HVGs were calculated, and used as input to construct an SNN-graph (k = 31) across all single cells. These were then clustered into closely connected communities using the Walktrap algorithm (*Pons and Latapy, 2005*). Clusters were annotated based on the co-expression of TEC subtype marker genes (*Figure 6—figure supplements 2* and *3*). Droplet single cells were visualised in reduced dimension with the first 20 PCs as input using uniform manifold approximation and projection (UMAP) (*McInnes et al., 2018*), with k = 31 nearest neighbours and a minimum distance = 0.3.

### Diffusion map and pseudotime inference

Diffusion maps and diffusion pseudotime trajectories were constructed using a matrix of log-transformed size-factor normalised gene expression values across single cells as input, with highly variable genes to define the diffusion components, implemented in the Bioconductor package *destiny* (*Angerer et al., 2016*; *Haghverdi et al., 2016*). Diffusion maps used k = 21 and the first 20 principal components as the input. Diffusion pseudotime distances were computed from an index cell defined in each analysis as the lowest position on DC1, representing the apex of bifurcating lineages. Density along pseudotime was calculated using a gaussian kernel density estimator with a fixed number of points proportional to 1% of cells ordered by diffusion pseudotime distance (DPT).

### Age-dependent cluster abundance modelling

Numbers of each TEC subtype or cluster were counted per replicate and at each age. Cell counts were modelled using a linear negative binomial model, with the total number of cells captured per replicate as a model weight, implemented in the Bioconductor package *edgeR* (*McCarthy et al., 2012*; *Robinson et al., 2010*). Statistically significant age-dependent changes were tested in these models using an empirical Bayes quasi-likelihood F-test (*Chen et al., 2016*).

### Age-dependent differential pseudotime state abundance

For the ZsGreen experiment, we assigned single-cells to differentiation states using either a 2- (Medulla branch 2) or 3-component (Medulla branch 1) gaussian mixture model (implemented in the R package *mclust*) fit to diffusion pseudotime distance (DPT) for cells in separate medulla lineages. Within each lineage and state, we then fit a Poisson GLM to the total number of cells in each replicate, and tested for linear age-dependent differences in abundance. Statistically significant differences were defined by a p-value$\leq$0.01 after a Bonferroni multiple testing correction.

### Tissue and tissue restricted antigen gene definition

Tissue restricted antigen (TRA) genes were defined based on the specificity of their expression across a broad range of mouse tissues using the FANTOM5 cap analysis of gene expression with sequencing (CAGE-seq) data that are publicly available (http://fantom.gsc.riken.jp/data/). Tissue samples were grouped into 27 broad groups based on the annotation data (*Supplementary file 1*-table 8). For each protein-coding gene (based on Ensembl identifier), the per-tissue expression level was defined as the maximum run length encoding (RLE) normalised expression level. For genes with multiple transcriptional start sites, the mean RLE expression across isoforms was first taken. The specificity of tissue expression for each gene across tissues (n) was then calculated using the tau-index ($\tau$) (*Yanai et al., 2005*):

$$\tau = \frac{\sum(1-\widehat{x})}{n-1}$$

$$\widehat{x} = \frac{x_i}{max(x_i)}$$

Genes with $\tau \geq 0.8$ were defined as TRAs, whilst those with $\tau \leq 0.4$ were defined as constitutively expressed; all remaining genes were given the classification 'miscellaneous'. Each TRA gene was assigned to one tissue, the one in which it was maximally expressed. *Aire*-dependent and -independent genes were defined using the classification from *Sansom et al., 2014*.

### Age-dependent tissue-representation modelling

The age-dependence of tissue-representation across single mTEC was tested using a negative binomial linear model. Specifically for each single mTEC the number of TRA genes with log expression >0 was counted within each assigned tissue (see above). These single-cell tissue counts were aggregated across single mTEC at each time point, and for each replicate mouse, to yield 'tissue counts'. Aggregated 'tissue counts' were then used as the dependent variable in a negative binomial linear model implemented in the Bioconductor package *edgeR*. Statistically significant age-dependent changes were defined at an FDR of 1%.

## Differential gene expression testing

All differential gene expression testings were performed in a linear model framework, implemented in the Bioconductor package *limma*. To test for age-dependent gene expression changes, log-normalised gene expression values for each gene was regressed on log2(age) and adjusted for sequencing depth for each single cell using deconvolution size factors estimated using *scran*.

## Gene signature and functional enrichment testing

Marker genes or differentially expressed genes (throughout ageing) were tested to identify enriched pathways, specifically those from MSigDB hallmark genesets or Reactome pathways. Marker genes were identified as those genes with a fourfold enrichment in the subtype relative to all other subtypes (adjusted p<0.01). MSigDB hallmark (*Liberzon et al., 2015*; *Subramanian et al., 2005*) and Reactome pathway (*Fabregat et al., 2018*) enrichments for markers of each subtype were computed using the clusterProfiler package (*Yu et al., 2012*). For age-dependent differentially expressed genes, gene set enrichment analysis was used (GSEA) to identify enriched MSigDB hallmark genesets. These results were categorised based on the expected change in expression due to ageing across multiple tissues and species (*Benayoun et al., 2019*).

## Age-dependent modelling of thymocyte negative selection

Age-dependent variation in thymocyte negative selection was modelled using a negative binomial GLM implemented in the Bioconductor package *edgeR*. Cell counts were regressed on age, using the input parent population for each replicate as a model offset to control for variation in the preceding selected population. Across populations, multiple testing was accounted for using the false discovery rate procedure (*Benjamini and Hochberg, 1995*), where a statistically significant relationship with age was set at 1%.

## Code and data availability

All code used to process data and perform analyses is available from https://github.com/WTSA-Homunculus/Ageing2019 (*Baran-Gale, 2020*; copy archived at https://github.com/elifesciences-publications/Ageing2019). All sequence data, counts matrices and meta-data are available from ArrayExpress with accession numbers E-MTAB-8560 (ageing thymus) and E-MTAB-8737 (lineage traced thymus). TCR sequencing data is available from SRA (PRJNA551022).

# Acknowledgements

We gratefully acknowledge the Chan Zuckerberg Biohub for support and for sequencing, and members of the Tabula Muris Consortium for technical assistance. CPP is funded by the MRC (MC_UU_00007/15). CPP, GH, JB and MDM were supported by the Wellcome Trust (grant 105045/Z/14/Z). JCM was supported by core funding from the European Molecular Biology Laboratory and from Cancer Research UK (award number 17197). GH and ICA was supported by the Swiss National Science Foundation (grant numbers IZLJZ3_171050 and 310030_184672). AEH was supported by a Clinical Lectureship from the NIHR. FD was supported by the Wellcome Trust [109032/Z/15/Z].

# Additional information

### Competing interests

Chris P Ponting: Reviewing editor, *eLife*. The other authors declare that no competing interests exist.

### Funding

| Funder | Grant reference number | Author |
| --- | --- | --- |
| Medical Research Council | MC_UU_00007/15 | Chris P Ponting |
| Wellcome | 105045/Z/14/Z | Jeanette Baran-Gale<br>Michael D Morgan<br>Georg A Holländer |

| Wellcome | 109032/Z/15/Z | Fatima Dhalla |
|---|---|---|
| Swiss National Science Foundation | IZLJZ3_171050 | Irene Calvo-Asensio<br>Georg A Holländer |
| Swiss National Science Foundation | 310030_184672 | Irene Calvo-Asensio<br>Georg A Holländer |
| Chan Zuckerberg Biohub | | Ashley Maynard<br>Steven Chen<br>Foad Green<br>Rene V Sit<br>Norma F Neff<br>Spyros Darmanis<br>Weilun Tan<br>Andy P May |
| European Molecular Biology Laboratory | 17197 | John C Marioni |
| National Institute for Health Research | | Adam E Handel |

The funders had no role in study design, data collection and interpretation, or the decision to submit the work for publication.

## Author contributions

Jeanette Baran-Gale, Michael D Morgan, Conceptualization, Data curation, Formal analysis, Validation, Visualization, Writing - original draft, Writing - review and editing; Stefano Maio, Irene Calvo-Asensio, Investigation, Methodology, Writing - review and editing; Fatima Dhalla, Investigation, Writing - original draft, Writing - review and editing; Mary E Deadman, Investigation, Project administration, Writing - review and editing; Adam E Handel, Data curation, Formal analysis, Writing - review and editing; Ashley Maynard, Steven Chen, Foad Green, Rene V Sit, Norma F Neff, Spyros Darmanis, Weilun Tan, Resources; Andy P May, Resources, Supervision; John C Marioni, Conceptualization, Supervision, Writing - original draft, Writing - review and editing; Chris P Ponting, Conceptualization, Supervision, Methodology, Writing - original draft, Writing - review and editing; Georg A Holländer, Conceptualization, Supervision, Funding acquisition, Methodology, Writing - original draft, Writing - review and editing

## Author ORCIDs

Jeanette Baran-Gale  https://orcid.org/0000-0002-8779-328X
Michael D Morgan  https://orcid.org/0000-0003-0757-0711
Irene Calvo-Asensio  http://orcid.org/0000-0002-9356-6008
Adam E Handel  http://orcid.org/0000-0001-8385-6346
John C Marioni  http://orcid.org/0000-0001-9092-0852
Chris P Ponting  https://orcid.org/0000-0003-0202-7816
Georg A Holländer  https://orcid.org/0000-0002-8790-0874

## Ethics

Animal experimentation: All mice were maintained under specific pathogen-free conditions and experiments were approved by the University of Oxford Clinical Medicine Ethical Review Committee and licensed under the Animals Scientific Procedures Act of the UK Home Office or Swiss cantonal and federal regulations and permissions (Permit *2321), depending where the mice were housed.

## Decision letter and Author response

Decision letter https://doi.org/10.7554/eLife.56221.sa1
Author response https://doi.org/10.7554/eLife.56221.sa2

# Additional files

## Supplementary files

• Supplementary file 1. Tables listing relevant experiment related information. (Table 1) Numbers of TEC isolated in the single-cell experiment. (Table 2) Marker genes for each subtype of TEC. (Table 3) Tests conducted for marker proteins in TEC. (Table 4) Single-cell defined TEC subtypes and known concordant phenotypes. (Table 5) Details of antibodies used in flow cytometry staining panels to identify TEC and thymocytes undergoing negative selection. (Table 6) ADT primers for Droplet sequencing. (Table 7) HTO primers for Droplet sequencing. (Table 8) Tissue-specific gene classification via FANTOM5 Cage-Seq:Tissue samples were grouped into 27 broad groups based on the annotation data.

• Transparent reporting form

## Data availability

Sequencing data have been deposited at ArrayExpress with accession numbers E-MTAB-8560 (ageing thymus) and E-MTAB-8737 (lineage traced thymus) or from SRA with accession number PRJNA551022 (TCR sequencing data).

The following datasets were generated:

| Author(s) | Year | Dataset title | Dataset URL | Database and Identifier |
|---|---|---|---|---|
| Baran-Gale J, Morgan MD, Maio S, Dhalla F, Calvo-Asensio I, Deadman ME, Handel AE, Maynard A, Chen S, Green F, Sit RV, Neff NF, Darmanis S, Tan W, May AP, Marioni JC, Ponting CP, Holländer GA | 2020 | Single-cell RNA-sequencing of mouse thymic epithelial cells across the first year of life | https://www.ebi.ac.uk/arrayexpress/experiments/E-MTAB-8560 | ArrayExpress, E-MTAB-8560 |
| Baran-Gale J, Morgan MD, Maio S, Dhalla F, Calvo-Asensio I, Deadman ME, Handel AE, Marioni JC, Ponting CP, Holländer GA | 2020 | Charting the age-altered thymic epithelial cell differentiation by lineage tracing from a beta 5-t expressing TEC progenitor | https://www.ebi.ac.uk/arrayexpress/experiments/E-MTAB-8737 | ArrayExpress, E-MTAB-8737 |
| Baran-Gale J, Morgan MD, Maio S, Dhalla F, Calvo-Asensio I, Deadman ME, Handel AE, Marioni JC, Ponting CP, Holländer GA | 2020 | TCR-seq of negatively selected T-cells | https://www.ncbi.nlm.nih.gov/bioproject/?term=PRJNA551022 | NCBI BioProject, PRJNA551022 |

The following previously published datasets were used:

| Author(s) | Year | Dataset title | Dataset URL | Database and Identifier |
|---|---|---|---|---|
| Bornstein C, Nevo S, Giladi A, Kadouri N, Pouzolles M, Gerbe F, David E, Machado A, Chuprin A, Tóth B, Goldberg B, Itzkovitz S, Taylor N, Jay P, Zimmermann VS, | 2018 | Large-scale single cell mapping of the thymic stroma identifies a new thymic epithelial cell lineage | https://www.ncbi.nlm.nih.gov/geo/query/acc.cgi?acc=GSE103970 | NCBI Gene Expression Omnibus, GSE103970 |

Abramson J, Amit I

| Park J, Botting RA, Conde CD, Popescu D, Lavaert M, Kunz DJ, Goh I, Stephenson E, Ragazzini R, Tuck E, Wilbrey-Clark A, Roberts K, Kedlian VR, Ferdinand JR, He X, Webb S, Maunder D, Vandamme N, Mahbubani KT, Polanski K, Mamanova L, Bolt L, Crossland D, Rita F, Fuller A, Filby A, Dixon RGD, Saeb-Parsy K, Lisgo S, Henderson D, Vento-Tormo R, Bayraktar OA, Barker RA, Meyer KB, Saeys Y, Bonfanti P, Behjati S, Clatworthy MR, Taghon T, Haniffa M, Teichmann SA | 2020 | A cell atlas of human thymic development defines T cell repertoire formation | https://www.ebi.ac.uk/arrayexpress/experiments/E-MTAB-8581/ | ArrayExpress, E-MTAB-8581 |

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
