## [Decision Letter]

**Acceptance summary:**

The manuscript provides insights into the diversity of thymic epithelial cell types by identifying likely precursor populations, constructing precursor-product relationships, and documenting what may represent a block in differentiation with age. It also provides useful characterization of changes in thymocytes with age, as a consequence of altered function of thymic epithelial cells. This revised version clarifies several important points, including the comparison with 2 public datasets (Park et al. and Bornstein et al) and the addition of a table with a list of marker genes for each TEC cluster, that further support the identification of 9 TEC subsets.

**Decision letter after peer review:**

Thank you for submitting your article "Ageing compromises mouse thymus function and remodels epithelial cell differentiation" for consideration by *eLife*. Your article has been reviewed by four peer reviewers, one of whom is a member of our Board of Reviewing Editors, and the evaluation has been overseen by and Tadatsugu Taniguchi as the Senior Editor. The following individual involved in review of your submission has agreed to reveal their identity: Magali Irla (Reviewer #3).

The reviewers have discussed the reviews with one another and the Reviewing Editor has drafted this decision to help you prepare a revised submission.

As the editors have judged that your manuscript is of interest, but as described below that additional experiments are required before it is published, we would like to draw your attention to changes in our revision policy that we have made in response to COVID-19 (https://elifesciences.org/articles/57162). First, because many researchers have temporarily lost access to the labs, we will give authors as much time as they need to submit revised manuscripts. We are also offering, if you choose, to post the manuscript to bioRxiv (if it is not already there) along with this decision letter and a formal designation that the manuscript is 'in revision at *eLife*'. Please let us know if you would like to pursue this option. (If your work is more suitable for medRxiv, you will need to post the preprint yourself, as the mechanisms for us to do so are still in development.)

Summary:

The authors use single cell analyzes of thymic epithelial cells (TECs) of mice at different ages to explore the mechanisms of thymic involution and its impact on T cell development. Thymic involution and change in the immune response with aging have been well documented, but the underlying mechanisms are poorly understood, due in part to a lack of information about TEC heterogeneity and dynamics at different ages of thymus development and involution. The authors therefore address an important question. Although it was known that ageing alters TEC composition and transcriptional profile (Gray et al., 2006; Ki et al., 2014), the originality of this study relies on using scRNA-seq during ageing, providing a substantial amount of novel data on TEC biology. The current study identifies potentially interesting new subsets, including one termed "intertypical TECs" that expresses features of both cTEC and mTECs. However, the reviewers had a number of significant concerns about the presentation and interpretation of the data as outlined below.

Essential revisions:

1) Although the study features some novel TEC populations (e.g. structural TECs, or n-TECs), the authors provide just two markers per cluster, without showing how homogeneously these markers are expressed within the specific cluster. Moreover, the study lacks validation of markers defining particular cell population by independent approaches. Similar lack of validation also applies for the main conclusions regarding changes in composition of TECs.

2) How do the authors reconcile the observation that they found 9 TEC subsets while Bornstein et al., 2018 only identified 5 TEC subsets (4 mTEC subsets and 1 cTEC subset) by scRNA-seq? Figure S3 compares Bornstein scRNA-seq data to the TEC subsets identified in the present study in different conditions (embryonic stages, Aireko, Pou2f2ko mice). The important data should be more clearly presented and discussed in the Results section. The legend for this Figure is confusing.

3) More characterization is needed for the novel "intertypical TEC" population described here. For example, do single TEC really do co-express *Psmb11* and *Ccl21a*, or is this TEC population merely a heterogenous cluster of cTEC and mTEC1? How do the intertypical TECs differ from previously described Ccl21+ cells or mTEC1 cells or jTECs, and what is their relationship with proliferating mTECs and mature mTECs? Figure 1 and their names suggest that intertypical TECs might give rise to proliferating mTECs and that they give rise to mature mTECs, however lineage tracing experiment shows intertypical and mature population to be highly ZsGreen positive, while proliferating mTECs are almost completely negative. How is the progenitor subset defined?

4) Potentially key data in the present study are from the lineage-tracing experiments in Figure 3 and 4, but the results here are difficult to interpret. Figure 3 isolates cells labeled in Psmb11-constrained Cre mice (termed 3xtg) after 48h, subjects them to scRNA-seq, and concludes that many labeled cells are what the authors call intertypical cells, including some cTEC and also cells termed by others as mTEC I that express *Ccl21a*. Figure 4 studies thymic epithelial cell populations 4 weeks after labeling, at different ages. However, labeling of populations at different ages is not assessed at early times, but only after 4 weeks. The authors do show expression of *Psmb11* in the scRNA-seq, but this does not establish that similar populations were labeled and at equivalent efficiency. This is the biggest omission in this work, that if rectified would greatly improve the work.

5) Additionally, the results are sometimes presented in a cryptic fashion, making it difficult to evaluate key aspects of the study. Several supplementary figures are of poor resolution (S1A, S1C, S5, S8B, S8C, S8D, S12C-F). They contain important data that cannot be properly evaluated. Moreover, there is an enormous amount of information – some of it important – in the supplemental information, that would benefit from being included in the main paper. This includes details of TRAs whose expression changes with age, and some important information for the lineage tracing experiments.

6) In its current form, the manuscript does not provide strong evidence that any of these described populations are progenitors, and the conclusion that differentiation is compromised with age, is therefore weak. The authors should be encouraged to discuss alternative interpretations of their data.

---

## [Author Response]

Essential revisions:1) Although the study features some novel TEC populations (e.g. structural TECs, or n-TECs), the authors provide just two markers per cluster, without showing how homogeneously these markers are expressed within the specific cluster. Moreover, the study lacks validation of markers defining particular cell population by independent approaches. Similar lack of validation also applies for the main conclusions regarding changes in composition of TECs.

The revised manuscript places greater emphasis on Supplementary file 1-table 2 (from the original manuscript) containing the full list of marker genes for each cluster, including *p*-values. The main text of the original submission necessarily can discuss only a subset of these. We have now amended this table to demonstrate the homogeneity of marker gene expression. It now includes statistics related to marker gene expression both within and outside of the cluster (specifically, mean expression and percentage non-zero expression within/outside of the cluster).

With regard to independent validation of markers, we now present 3 lines of evidence that fully support (a) our identification of 9 TEC populations, and (b) genes whose expression is enriched within these populations:

1) In two revised Figures (Figure 2D and Figure 2—figure supplement 2) we now provide independent replication of our TEC types in two separate mouse data sets, namely those of Bornstein et al. (Bornstein et al., 2018) and Park et al. (Park et al., 2020) (also relevant to (2) below). These studies were performed independently in different institutes, involving mice of different ages and genders, and were analysed using different computational workflows. Both studies also used single-cell mRNA sequencing to study TEC heterogeneity owing to this technique’s unrivalled high throughput and predictive power.

2) Single antibody experiments were unable to discriminate the 9 TEC populations for 3 reasons: (i) These cells’ marker proteins are commonly extracellular or intracellular, rather than being cell-surface. For example, the Intertypical TEC subtype marker *CCL21a* is secreted resulting in a diffuse staining pattern by IHC. (ii) Marker protein levels are not always expressed in an age-independent manner. For example expression of Ly51, the well-established canonical marker of cTEC, varies considerably across the life-course, yielding sorted populations that contain both mature cTEC and Intertypical TEC, increasingly so in older mice. (iii) The unavailability of high-quality and/or informative reagents. We tested 18 protein-assay combinations (14 antibodies targeting 13 proteins by either histology and/or flow cytometry), of which 10 were technically successful (i.e. provided a signal above background noise levels). These proteins were: for histology, CD177 (2 different clones), Lifr, Cytokeratin5, Cytokeratin8 and *CCL21a*, and for flow cytometry, CD177, L1cam, *Mmp2* and CD142. We further tested antibodies against CD52, Tagln, Dcamkl1, Itgb4 and CD271, which did not yield signals above background staining. To assist the TEC research community we provide full details of these antibodies and assays in Supplementary file 1-table 3.

3) The compositional changes we observe are concordant with a previous report (Rode, et al. J Immunol 2015). In pre-pubescent mice, cTEC are initially the most abundant epithelial cell subpopulation. At approximately 4 weeks of age the cellularity of mTEC is the largest and thereafter their number (and consequently the size of the medulla) declines, whereas the cTEC scaffold expands in parallel. In this manuscript, we considerably extend this knowledge by demonstrating that cTEC in 1-week old mice are transcriptomically distinct from cTEC in older mice. Moreover, we show that the cytometrically isolated cTEC includes perinatal cTEC, mature cTEC and Intertypical TEC, a composition that changes with age. The division of Intertypical TEC across commonly used flow cytometric gates further illustrates the challenges in validating TEC subpopulation shifts using flow cytometry and a limited set of informative cell surface markers.

For these 3 reasons, we believe that high-dimensional single-cell methods are the only currently-available approach able to definitively resolve TEC heterogeneity.

The revised manuscript now directly addresses these points:

– Additional information concerning marker gene heterogeneity is present in Supplementary file 1-table 2 (as described above).

– New Supplementary file 1-table 3 details our FACS/Histology assays that sought to validate markers.

– The main text and Figure 2 have been modified to further clarify these issues:

“Our analysis revealed 9 TEC subtypes (Figure 2b,c), thus providing a greater richness of epithelial states than previously appreciated (Bornstein et al., 2018, Park et al., 2020) (Figure 2D, Figure 2—figure supplement 2) and a greater diversity than the 4 phenotypes cytometrically defined and selected in this study (Figure 2—figure supplement 3A-B). […] Hereafter, for clarity, we refer to transcriptomically-defined TEC clusters as subtypes and cytometrically-specified TEC as subpopulations.”

2) How do the authors reconcile the observation that they found 9 TEC subsets while Bornstein et al., 2018 only identified 5 TEC subsets (4 mTEC subsets and 1 cTEC subset) by scRNA-seq? Figure S3 compares Bornstein scRNA-seq data to the TEC subsets identified in the present study in different conditions (embryonic stages, Aireko, Pou2f2ko mice). The important data should be more clearly presented and discussed in the Results section. The legend for this Figure is confusing.

We are grateful to the reviewers for highlighting this important distinction between the two studies. We performed an independent clustering on data using, for consistency, the identical approach that we had taken with our own data. (Note that we could not take advantage of the identity of TEC subsets in the Bornstein et al. dataset as this information is not publicly available.) This independent clustering, which was described in our original submission, revealed using the Bornstein et al. dataset the presence of 9 TEC clusters that were highly concordant with the findings from our own clustering of TEC across the mouse life course (data from this study).

Our most recent (i.e. new) analysis demonstrates the consistent and independent replication of these 9 TEC subtypes now across 3 mouse data sets: wild-type epithelial cells from Bornstein et al., epithelial cells from Park et al., and our TEC data from this mouse ageing study. More specifically, we trained a k-nearest neighbour classifier using either the mouse TEC from Park et al. or the mouse TEC in our ageing study, and used these to classify the single TEC from Bornstein et al. The results of this cross-classification analysis are presented in two new figures (Figure 2D and Figure 2—figure supplement 2), and data complements our previous Table 1 (now Supplementary file 1-table 4). This latest analysis demonstrates an independent replication of the TEC subtypes across 3 data sets, and further provides a mapping between the different naming schema (new Figure 2 —figure supplement 2).

Furthermore, we note that the higher temporal resolution of our ageing time course has empowered the identification of the Perinatal cTEC subtype. We classified a small number of these cells within both the Bornstein et al. data from 6 day old mice (n=5), and the young TEC from Park et al. (n=4), both sets of which are disproportionately derived from 6 day old mice, as shown in Author response image 1 (Left-Bornstein et al., Right-Park et al.):

Finally, we identified 2 rare populations of mouse TEC: sTEC and nTEC. nTEC have also been described recently by Park et al., 2020.These new analyses are now presented in the revised manuscript:

– Figure 2D and Figure 2—figure supplement 2 include river plots that reconcile the mapping between TEC identities described by independent single cell TEC studies.

– Text as quoted under essential revision point 1) above.

– New text:

“With single-cell transcriptomics we identified 9 TEC subtypes, 4 of which were previously undescribed (Supplementary file 1-table 4). […] Similarly, our data from older mice demonstrates that Intertypical TEC (referred to as mTEC I in (Bornstein et al., 2018; Park et al., 2020)) are composed of both cytometrically sorted cTEC and mTEC^lo^ populations.”

3) More characterization is needed for the novel "intertypical TEC" population described here. For example, do single TEC really do co-express Psmb11 and Ccl21a, or is this TEC population merely a heterogenous cluster of cTEC and mTEC1? How do the intertypical TECs differ from previously described Ccl21+ cells or mTEC1 cells or jTECs, and what is their relationship with proliferating mTECs and mature mTECs?

We have now updated the description of Intertypical TEC to clarify their gene markers and their relationship with previously published subpopulations including mTEC I and jTEC (Supplementary file 1-table 4: previously Table 1). Accordingly, Intertypical TEC can be described as *Ccl21a+Ly6a+Pdpn+Plet1+* as listed in Supplementary file 1-table 4 (previously Table 1). We show the correspondence between Intertypical TEC and mTEC I in River Plots (new Figure 2D and new Figure 2—figure supplement 3). From these it should be evident that this TEC population is not a heterogeneous mixture of cTEC and mTEC I.

Regarding the co-expression of *Psmb11* and *Ccl21a*, these genes are co-expressed in a subset of Intertypical TEC (Author response image 2). It is important to note that β-5t-marked cells do not concurrently express *Psmb11*. This arises from differing dynamics between the cortical and medullary lineages, whereby cTEC-committed cells consistently up-regulate *Psmb11*, which is sustained in their mature state (Figure 7—figure supplement 2). In contrast, cells committed to the medullary lineage pass through a transient state, that includes the expression of *Psmb11*, which is then down-regulated (Figure 7E). Thus, we do not expect a strong constitutive co-expression of both *Psmb11* and *Ccl21a* as a marker of a distinct TEC population.

**Author response image 2. respfig2:** Co-expression of *Psmb11* and *Ccl21a* in single TEC. Expression in single TEC of (a) *Psmb11* , (b) *Ccl21a* , (c) both (*Psmb11*+/*Ccl21a*+; blue), and (d) expression levels of *Psmb11* vs. *Ccl21a*, with TEC subtype indicated by colour. TEC with no detected expression of the relevant gene are coloured in grey (a-b).

Figure 1 and their names suggest that intertypical TECs might give rise to proliferating mTECs and that they give rise to mature mTECs, however lineage tracing experiment shows intertypical and mature population to be highly ZsGreen positive, while proliferating mTECs are almost completely negative. How is the progenitor subset defined?

Our lineage tracing experiments indicate that the progression of mTEC differentiation proceeds from medulla-biased Intertypical TEC through a proliferative state to mature mTEC (Figure 5E, Figure 6C, Figure 7, Figure 7—figure supplement 1). We apologise that our original submission unintentionally misled the reviewers: proliferating mTECs are not “almost completely [ZsGreen] negative”. We now further clarify these issues in the revised manuscript, specifically by improving Figures 6 and 7 and associated supplemental Figures. These show results from the larger lineage tracing with single-cell RNA-sequencing analysis and demonstrate that whilst the labelling of Proliferating TEC declines modestly with age, approximately 60% of cells in this group remained labelled 4 weeks after doxycycline treatment (Figure 6E).

This new lineage tracing experiment demonstrated that Intertypical TEC derive from both β5t-positive and -negative precursors, and that Intertypical TEC represent a heterogeneous population expressing varying levels of *Pdpn*, *Ccl21a* and *Psmb11* (Figure 7E, Figure 6—figure supplement 5, Figure 7—figure supplement 3). We expect that these cells represent common states in TEC differentiation, rather than discrete cell types in the conventional sense, namely separate thymic epithelial subtypes that represent terminally differentiated TEC. Moreover, it is evident that Intertypical TEC sub-clusters represent lineage-biased states: for example, Intertypical TEC-3 express more cTEC-like markers and Intertypical TEC-1 and -2 express more mTEC-like markers (Figure 6—figure supplement 5). For the subsequent differentiation trajectory analysis we inferred that the Intertypical TEC-4 population as being the progenitor group based on: i) enrichment of *Pdpn*+ and *Ly6a+* (Figure 7E and Figure 6—figure supplement 5 and Figure 7—figure supplement 3), after which there is transient expression of *Psmb11*+ (Figure 7E) – these genes have previously been described as markers of TEC progenitors, and, ii) their position at the apex of the bifurcation between cTEC and mTEC lineages (Figure 7—figure supplements 1 and 2).

Our revised manuscript now contains far greater detail in contrast to our initial description of the Intertypical TEC population:

“In contrast, Intertypical TEC were present in both cortical and medullary subpopulations, and expressed gene markers associated with a progenitor-like TEC^lo^ phenotype (Supplementary file 1-table 4). […] Together, these results suggest: (i) that previous classifications annotated a heterogeneous population containing Intertypical TEC, and (ii) that Intertypical TEC might provide the progenitors of mature mTEC found at high density at the CMJ (Onder et al., 2015, Michel et al., 2017, Ulyanchenko et al., 2016, Lepletier et al., 2019, Mayer et al., 2016; Supplementary file 1-table 4).”

4) Potentially key data in the present study are from the lineage-tracing experiments in Figure 3 and 4, but the results here are difficult to interpret. Figure 3 isolates cells labeled in Psmb11-constrained Cre mice (termed 3xtg) after 48h, subjects them to scRNA-seq, and concludes that many labeled cells are what the authors call intertypical cells, including some cTEC and also cells termed by others as mTEC I that express Ccl21a. Figure 4 studies thymic epithelial cell populations 4 weeks after labeling, at different ages. However, labeling of populations at different ages is not assessed at early times, but only after 4 weeks. The authors do show expression of Psmb11 in the scRNA-seq, but this does not establish that similar populations were labeled and at equivalent efficiency. This is the biggest omission in this work, that if rectified would greatly improve the work.

Thank you for these very helpful comments. In the revised submission, we strengthen the case that the change in TEC differentiation dynamics is a consequence of ageing by providing additional information on: 1) the differences in labelled cells across all EpCAM+ cTEC and mTEC following either a 2 or 28 day chase period after doxycycline treatment, and 2) the dynamics of labelling as a function of age (Figure 6A and Figure 6—figure supplement 1). The majority of (ZsG+) labelled TEC after 48hours at all ages are cTEC (Author response image 3), in line with the constitutive expression of *Psmb11* in this TEC compartment. Notably ~60% of all cTEC are labelled after 2 days post dox-treatment in 1 week old mice (top), whilst the equivalent mTEC is close to 0%. The percentage of cells labelled in the cTEC then drops quickly to ~25% in 4 week old mice (middle), and to 25-40% in 16 week old mice (bottom), in accordance with the observed fluctuations in cTEC numbers in the thymus and the switch from perinatal to mature cTEC (Figure 3B). Concerning the mTEC of mice treated at 1 week, after 28 days ~25% of mTEC are labelled, which is equivalent to the number of labelled cTEC, and suggests a progressive accumulation of label in the mTEC compartment that is specific to the first 28 days of mouse life. From 4 weeks onwards, the percentage of labelled mTEC remains consistently low, in line with our current and previous [Mayer et al.] conclusions that the efficiency or extent of contribution to mTEC differentiation from β5t-positive precursors that are physically present at the CMJ is reduced with age.

**Author response image 3. respfig3:** 

It is important to emphasise that these results are consistent across replicates (*n*=5 per experiment) and that they illustrate the highly dynamic nature of TEC differentiation. This latter point is now highlighted in Figure 6A, which shows the dynamics of ZsGreen labelling over age for the different chase periods. The initial high labelling of cTEC in 1 week old mice after 2 days, gives way to a more consistent labelling across older mice and the period of time used for the chase, findings that are again concordant with a constitutive expression of *Psmb11*.Furthermore, in Figure 6A we observe that the percentage of labelled mTEC 2 days post dox treatment increases monotonically (i.e. progressively in the same direction) from 1 to 4 to 16 weeks, consistent either with a subset of mTEC being derived continuously from a β5t-positive progenitor, and/or there being an increase in promiscuous expression of *Psmb11* amongst mature mTEC, and/or there being an accumulation of ZsGreen labelling with age.

Consequently, our revised manuscript now contains further details on lineage tracing (including Figure 6A and Figure 6—figure supplement 1) and the section entitled Ageing compromises the differentiation of Intertypical TEC into mature mTEC has been revised extensively:

“Our current and previous observations indicate that TEC differentiation is highly dynamic across the mouse life course (Figure 3B,C (Mayer et al., 2016)). […] The accumulation of ZsG+ cells 2 days after doxycycline treatment in 16 week old mice was not maintained 4 weeks later in either cTEC or mTEC, evident by an approximately 2-fold reduction in labelled TEC (Figure 6A).”

5) Additionally, the results are sometimes presented in a cryptic fashion, making it difficult to evaluate key aspects of the study. Several supplementary figures are of poor resolution (S1A, S1C, S5, S8B, S8C, S8D, S12C-F). They contain important data that cannot be properly evaluated. Moreover, there is an enormous amount of information – some of it important – in the supplemental information, that would benefit from being included in the main paper. This includes details of TRAs whose expression changes with age, and some important information for the lineage tracing experiments.

We apologise. All image quality issues have been resolved and new high-quality versions of these figures are now presented. In addition, and to highlight the decline in mTEC function, we have presented the results on age-dependent changes in mTEC TRA promiscuous gene expression in a new Figure 4 (previously supplementary figure 7).

6) In its current form, the manuscript does not provide strong evidence that any of these described populations are progenitors, and the conclusion that differentiation is compromised with age, is therefore weak. The authors should be encouraged to discuss alternative interpretations of their data.

Thank you for these comments. Accordingly, we have now revised our manuscript extensively to include lineage tracing information and new analyses whose results support our previous conclusions. Specifically: (1) We show that perinatal cTEC express *Psmb11*, a marker expressed in thymic progenitors (Figure 2—figure supplement 4). (2) Moreover, we demonstrate that mTEC arise from a medulla-biased Intertypical TEC pool, and both are derived from a β5t+ precursor TEC state (Figure 5d-e, 6c, 7a-d). (3) We have revised our previous Figure 4 (now new Figures 6 and 7) which now also contain associated lineage tracing data and a new pseudotime analysis. These figures show that mTEC are most likely derived from one or more Intertypical TEC states and these share a β5t+ common progenitor. Our available data suggests that this common progenitor may be enriched within the Intertypical TEC-4 population. Experimental confirmation of this hypothesis would require further lineage tracing from an entirely new transgenic model and thus is out of scope for this submission. Intriguingly, we also observed an accumulation of ZsG negative cells in a specific Intertypical TEC state (Figure 7—figure supplement 1) that likewise represents an mTEC precursor based on our pseudotime analysis. We believe that this could represent a second differentiation trajectory within the mTEC lineage or a population of very long-lived quiescent Intertypical TEC.

In response, to reflect additional complexities that our experiments suggest we now discuss alternative possible explanations:

“TEC progenitors have been described with distinctive molecular identities (e.g. β-5t expression) from the postnatal thymus where they dynamically expand and contribute to the mTEC scaffold (Bleul et al., 2006; Ucar et al., 2014; Ulyanchenko et al., 2016; Wong et al., 2014). […] If the model proposed by Lepletier et al. is correct, then TEC progenitors may be the architects of their own malfunction.”